# Rab11 endosomes and Pericentrin coordinate centrosome movement during pre-abscission in vivo

Nikhila Krishnan[1,3] ◉, Maxx Swoger[2,3], Lindsay I Rathbun[1,3], Peter J Fioramonti[1,3], Judy Freshour[1,3] ◉, Michael Bates[1,3], Alison E Patteson[2,3], Heidi Hehnly[1,3] ◉

**The last stage of cell division involves two daughter cells remaining interconnected by a cytokinetic bridge that is cleaved during abscission. Conserved between the zebrafish embryo and human cells, we found that the oldest centrosome moves in a Rab11-dependent manner towards the cytokinetic bridge sometimes followed by the youngest. Rab11-endosomes are organized in a Rab11-GTP dependent manner at the mother centriole during pre-abscission, with Rab11 endosomes at the oldest centrosome being more mobile compared with the youngest. The GTPase activity of Rab11 is necessary for the centrosome protein, Pericentrin, to be enriched at the centrosome. Reduction in Pericentrin expression or optogenetic disruption of Rab11-endosome function inhibited both centrosome movement towards the cytokinetic bridge and abscission, resulting in daughter cells prone to being binucleated and/or having supernumerary centrosomes. These studies suggest that Rab11-endosomes contribute to centrosome function during pre-abscission by regulating Pericentrin organization resulting in appropriate centrosome movement towards the cytokinetic bridge and subsequent abscission.**

## Introduction

During the onset of cell division, the centrosome duplicates to build a bipolar mitotic spindle, which is a macromolecular machine that ensures each daughter cell receives an equal complement of chromosomes. After chromosome separation, cytokinetic furrow formation, and the dismantling of the bipolar spindle, the daughter cell centrosomes migrate from one end of the cell towards the cytokinetic bridge in HeLa cells and were suggested to be required for cytokinetic bridge cleavage, referred to as abscission (Piel et al, 2001). Supporting this idea, human cells lacking centrioles, after centrinone treatment, can present with increased multinucleation, indicative of abscission defects (Wang et al, 2020). Bringing into

possible question the role of the centrosome movement towards the bridge during pre-abscission was a study that demonstrated in different mammalian cell lines a range of centriole motions during pre-abscission, with some directed towards the bridge and other motions being highly irregular (Jonsdottir et al, 2010). We propose that directed movement of the centrosome towards the cytokinetic bridge may be used during tissue morphogenesis (Hall & Hehnly, 2021), where the cytokinetic bridge and associated midbody can act as a symmetry breaking event to direct events at the apical membrane (Little & Dwyer, 2021; Farmer & Prekeris, 2022) such as cilia formation (Bernabé-Rubio et al, 2016). This would require the centrosome to migrate towards this site and then subsequently dock with the apical membrane to form a cilium. A model tissue that can be used to examine this is the ciliated organ of asymmetry known as Kupffer's vesicle (KV) in *Danio rerio* (zebrafish). KV morphogenesis first requires a group of migratory cells to divide and then self-assemble into a rosette-like structure. KV cells then position their cytokinetic bridges towards the rosette center before the rosette transitions into a cyst-like structure with a hollow fluid-filled lumen (Rathbun et al, 2020b). Our studies expand upon these ideas, where we find that centrosomes reorient towards the cytokinetic bridge at the center of the rosette. In addition, we identify a potential mechanism involving the small GTPase Rab11 in centrosome motility both in human cell culture and in zebrafish embryonic cells.

Rab11 and its associated membrane compartment, recycling endosomes (REs), localize to a specific sub-structure of the interphase centrosome, mother centriole appendages (Hehnly et al, 2012). This association was shown to help modulate cargo transport through the RE (Hehnly et al, 2012; Naslavsky & Caplan, 2020). REs are an endocytic compartment that accepts endocytosed cargo from the early endosomes and recycles it back to the plasma membrane (Welz et al, 2014). Owing to the nature of centriole duplication, the two daughter cell centrosomes are inherently asymmetric from one another with the oldest centrosome being enriched with centriole appendage proteins (Hung et al, 2016; Colicino et al, 2019) and more competent for cilia formation (Anderson & Stearns, 2009). When RE function is disrupted by

---

[1]Department of Biology, Syracuse University, Syracuse, NY, USA    [2]Department of Physics, Syracuse University, Physics Building, Syracuse, NY, USA    [3]BioInspired Institute, Syracuse University, Syracuse, NY, USA

Correspondence: hhehnly@syr.edu
Lindsay I Rathbun's present address is University of Rochester, Rochester, NY, USA.

depleting Rab11 in human cells in culture, centrosome function is potentially disrupted resulting in a loss of cilia formation during $G_0$ (Knödler et al, 2010; Westlake et al, 2011; Xie et al, 2019). Based on this relationship between Rab11-REs and the oldest of the centriole pair (the mother centriole) during interphase, we hypothesize that the centrosome requires an association with Rab11-REs during pre-abscission for its directed movement towards the cytokinetic bridge.

Many modes of centrosome movement have been documented (Vaughan & Dawe, 2011), of note is centrosome migration during *Drosophila melanogaster* neuroblast divisions (Rebollo et al, 2007; Rusan & Peifer, 2007; Lerit & Rusan, 2013; Hannaford et al, 2022 *Preprint*). In this context, the centrosome duplicates during S-phase where one centrosome migrates to the distal side of the cell to mature. This migratory process involves Pericentrin-like protein along with Polo Kinase (Lerit & Rusan, 2013). Our studies have expanded upon these findings where we identify that directed centrosome movement towards the cytokinetic bridge uses mechanisms involving both Pericentrin and Rab11-endosomes.

The RE compartment is involved in numerous cellular processes including, but not limited to, cilia formation, lumenogenesis, apical polarity formation, and abscission (Wilson et al, 2005; Bryant et al, 2010; Knödler et al, 2010; Westlake et al, 2011; Rathbun et al, 2020b). Rab11-associated REs transport with their associated cargo into the cytokinetic bridge (Montagnac et al, 2008) where they fuse at the cleavage furrow (Goss & Toomre, 2008). After furrow ingression, animal cells stay interconnected for some time by a narrow intercellular bridge that contains a proteinaceous structure known as the midbody (reviewed in Chen et al [2012]). These endosomes can fuse and potentially prime the membranes next to the midbody for an abscission event (Schiel et al, 2012). When depleting Rab11 using siRNAs (Wilson et al, 2005) or inhibiting the ability of Rab11-associated vesicles to transport into the bridge using optogenetics (Rathbun et al, 2020b), abscission failure occurs both in cell culture and in the zebrafish embryo. However, the relationship between Rab11-associated REs and the centrosome during this process has not been investigated. Here we investigate whether Rab11 can influence centrosome function and movement during pre-abscission.

# Results

### Differences in mitotic centrosome movement towards the cytokinetic bridge during pre-abscission between zebrafish embryos and human cells

We tested whether centrosome reorientation towards the cytokinetic bridge is a conserved process by comparing zebrafish embryo cells at 50% epiboly (3–5 hours post fertilization (hpf), Figs 1A and E and S1A), zebrafish KV cells (Fig 1B–E), and human HeLa cells (Figs 1E and S1B). Dividing zebrafish cells expressing two different centrosome markers, centrin-GFP (Zolessi et al, 2006; Rathbun et al, 2020a) or PLK1-mCherry (PLK1-mCh) (Rathbun et al, 2020b), were imaged (Figs 1A–D and S1A and Video 1) and compared with human (HeLa) cells expressing the centrosome markers DsRed-PACT (a C-terminal centrosome targeting domain taken from Pericentrin, [Vertii et al, 2015], Fig S1B) or centrin-GFP (Fig 1E, [Piel et al, 2001; Kuo et al, 2011]). Dividing cells are marked by expression of PLK1-mCh

during epiboly and KV development, which can mark the centrosome and cytokinetic midbody (Fig 1a', C, and D). Live cell Imaging was performed on both dividing epiboly cells and KV cells where we note at least one centrosome reorienting towards the cytokinetic bridge (Fig 1A, a', C, and D). Specifically, with KV dividing cells we find that the dividing cell within a rosette pinches its cytokinetic bridge towards the rosette center, and then begins to reorient at least one of the two centrosomes towards that site (Fig 1B–D).

Centrosome movement was quantified by monitoring live-cell acquisitions from embryos and human cells. We calculated whether both daughter cell centrosomes, only one daughter cell centrosome, or neither centrosome moved from the polar ends of the daughter cells across the cell centroid position (modeled in Fig S1D) towards the cytokinetic bridge (Fig S1C–E). We also determined the distance of the centrosome from the cytokinetic bridge in human cells by drawing a line from the centrosome to the midpoint of the cytokinetic bridge (modeled in Fig S1D). We identified in human cells that most cells moved both their daughter cell centrosomes towards the cytokinetic bridge (Fig 1E), and most of the centrosomes moved within a 4-$\mu$m distance from the bridge center (Fig S1E) with very few entering into the cytokinetic bridge itself (example of movements in Fig S1A and C). In zebrafish epiboly cells, KV cells, and human cells we find that in most cases at least one centrosome moves towards the cytokinetic bridge (Fig 1E). However, we do find a significant difference between human cells and zebrafish cells in which both centrosomes (human) or one centrosome (zebrafish) preferentially moves towards the bridge (Fig 1E). Unlike human cells where one centrosome moves in before the other (Fig S1B), we find that a significant majority of zebrafish pre-abscising cells during epiboly and KV development had only one centrosome move consistently towards the cytokinetic bridge before bridge abscission (Fig 1E). One possibility with KV cells is that the other centrosome moves up to the apical site after abscission. Because one centrosome moves towards the cytokinetic bridge before the other in human cells, and invariably only one centrosome moves towards the bridge in zebrafish pre-abscising cells, this suggests a potential asymmetry between the two centrosomes, such as centrosome age, contributing to this behavior.

### Mitotic centrosomes associate with Rab11 endosomes as they reorient towards the cytokinetic bridge

Rab11 could be a modulator of centrosome movement because its known to associate with mother centriole appendage proteins during interphase (Hehnly et al, 2012). However, its dynamic organization in relation to the centrosome during mitotic exit has not been specifically investigated. We examined Rab11 localization between human (HeLa) cells and dividing cells in the zebrafish embryo. We monitored spatial and temporal association of Rab11, and its effector protein, family of Rab11-interacting protein (FIP3), with the centrosome during pre-abscission in HeLa cells. The Rab11-effector protein FIP3 was tagged with GFP (Wilson et al, 2005) and expressed in HeLa cells with a centrosome marker, DsRed-PACT (Fig 2A). We found a population of REs at the centrosomes during anaphase and this population significantly increased as the cell progressed through cytokinesis and pre-abscission (Figs 2A and S2A and B and Video 2). A second population of REs then forms adjacent to the cytokinetic bridge following the formation of the centrosome

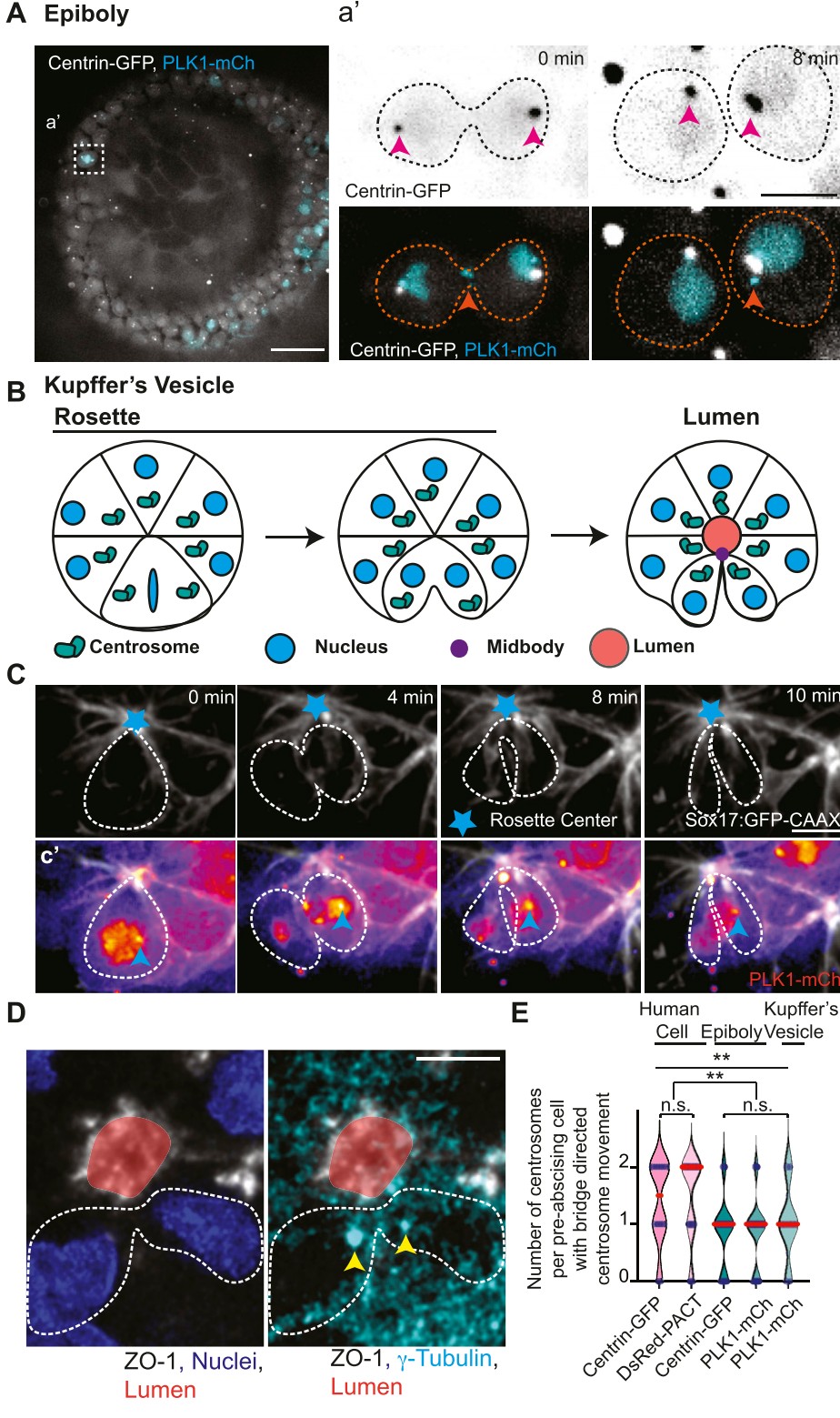

**Figure 1. Differences in mitotic centrosome movement towards the cytokinetic bridge during pre-abscission between zebrafish embryos and human cells.**

**(A)** Zebrafish embryo (5 h post fertilization) with centrin-GFP (gray) and PLK1-mCh (cyan). Scale bar, 50 $\mu$m. **(a')** Inset of dividing cell in (A). Time-lapse of centrin-GFP (inverted grays, top panel; grays, bottom panel) and PLK1-mCh (cyan). Video 1 Pink arrow, centrosome. Orange arrow, midbody. Dashed lines, cell boundaries. Scale bar, 10 $\mu$m. **(B)** Model depicting centrosome (green) movement towards the cytokinetic bridge in dividing cells within the Kupffer's vesicle (KV) during its development. Cyan, Nucleus. Purple, Midbody. Orange, Lumen. Dark lines, KV membranes. **(C)** Time-lapse of a dividing cell within the KV. KV cell membranes marked with Sox17:GFP-CAAX (gray). Cyan star, rosette center. Cyan arrow, centrosome. Dashed lines, cell boundaries. Scale Bar, 10 $\mu$m. **(c')** Dividing cell depicted with PLK1-mCh (fire LUT). Cyan arrow, centrosome. **(D)** A KV pre-abscising cell fixed and immunostained for ZO-1 (gray), $\gamma$-tubulin (cyan) and DNA (DAPI, blue). Yellow arrow, centrosome. Dashed lines, cell boundaries. Scale bar, 10 $\mu$m. **(E)** Number of centrosomes per pre-abscising cell with bridge directed centrosome movement calculated as both centrosomes (2 centrosomes), only one centrosome (1 centrosome) and neither centrosome (0 centrosomes) moved shown as a violin plot with median (orange) and quartiles (dark dotted lines). Two-tailed $t$ test between Centrin-GFP and DsRed-PACT in Human (HeLa) cells (pink background). n > 10 cells across n > 3 experiments n.s. not significant. One-way ANOVA across zebrafish epiboly cells and KV cells (green background), n.s. not significant. n > 10 cells across n > 2 embryos. One-way ANOVA, across all columns, **$P$ < 0.01. Two-tailed $t$ test between Human (HeLa) cells and zebrafish (Epiboly, KV) cells, **$P$ < 0.01. n-values, detailed statistical results in Table S1.

population, the centrosome then reorients with a population of REs to this second RE population (Figs 2A and E and S2A and Video 2). Another membrane organelle, the Golgi apparatus (labeled with MannII-mRuby2, [Lam et al, 2012]), is dispersed in small puncta during cytokinesis and starts to form two separate fragmented compartments next to the centrosome and the bridge (Fig S2C). This localization pattern is different from the REs that remain tightly organized at the centrosome and adjacent to the bridge (Fig S2A). To

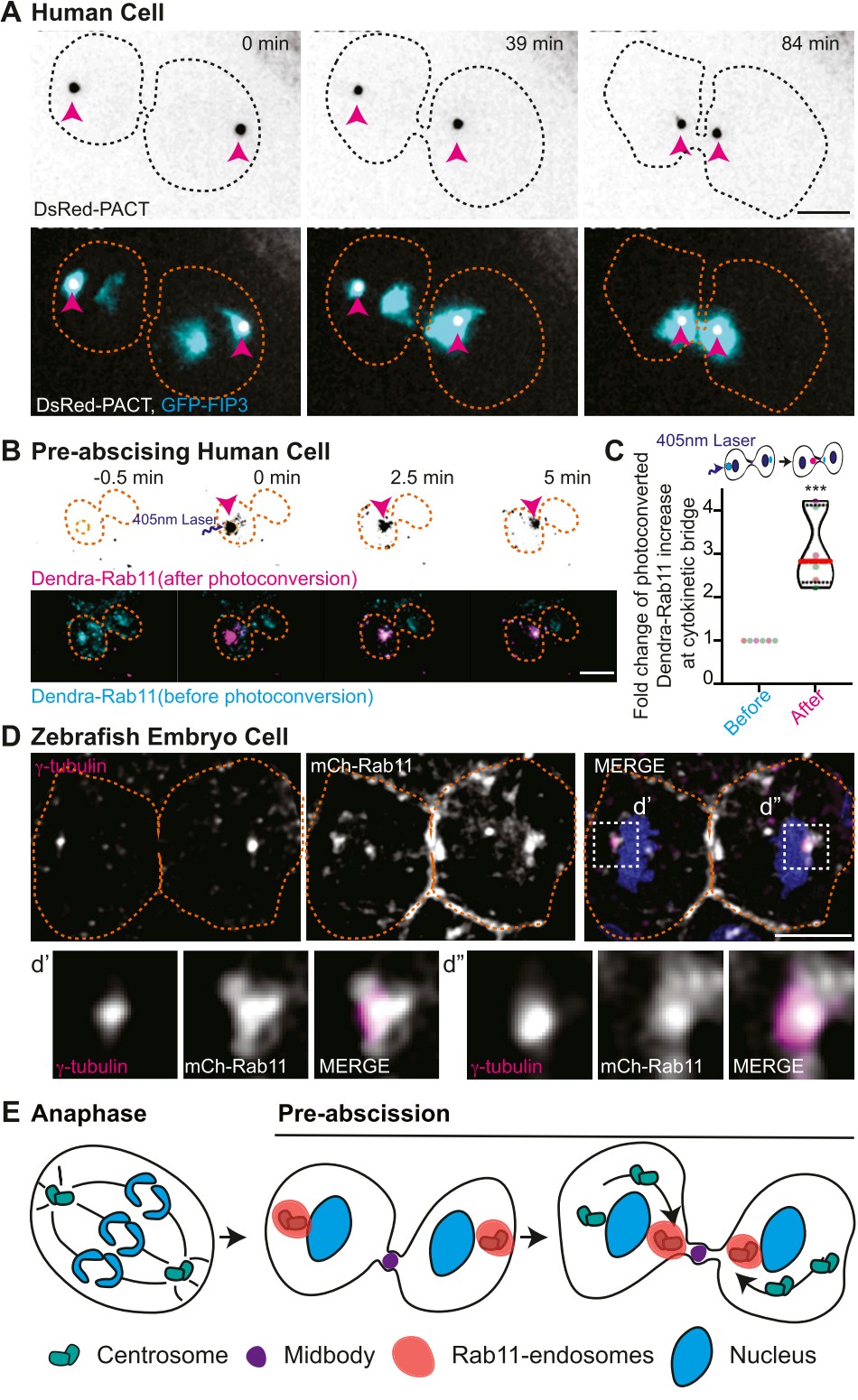

**Figure 2. Mitotic centrosomes associate with Rab11-endosomes as they reorient towards the cytokinetic bridge.**
**(A)** Time-lapse of pre-abscising human (HeLa) cell expressing DsRed-PACT (inverted grays, top panel; gray in merge, bottom panel) and GFP-FIP3 (cyan in merge, bottom panel). Video 2 Pink arrow, centrosome. Dashed lines, cell boundaries. Scale bar, 10 μm. **(B)** Time-lapse of a pre-abscising human (Hela) cell expressing Dendra-Rab11. Dendra-Rab11 is photoconverted from 507 nm emission (cyan in merge, bottom panel) to 573 nm emission (inverted grays, top panel; magenta in merge, bottom panel) using 405 nm light within an region of interest over the centrosome (pink arrow) at 0 min. Dashed lines, cell boundaries. Scale bar, 10 μm. **(C)** Violin plot with median (orange dashed line) and quartiles (black lines) depicting fold change of photoconverted Dendra-Rab11 (573 nm emission) increase at the cytokinetic bridge. Each dot in the plot represents a cell and the different colors depict separate experiments. n = 6 cells and n = 4 experiments. Two-tailed t test, ***P < 0.001. n-values and statistical results detailed in Table S1. **(D)** A zebrafish embryo pre-abscising cell expressing mCh-Rab11 (gray) fixed and immunolabeled for γ-tubulin (magenta) and DNA (DAPI, blue). Dashed lines, cell boundaries. Scale bar, 10 μm. Insets (d' and d''), 2X magnification. **(E)** Model depicting centrosomes (green) containing Rab11-endosomes (orange) reorienting towards the cytokinetic bridge with associated midbody (purple) during pre-abscission.

test if the second population of REs originated from the REs organized at the centrosome, a photo-convertible Dendra-Rab11 was used. The centrosome population of Dendra-Rab11a endosomes were photo-converted from a 507-nm emission to a 573-nm emission by placing a region of interest (ROI) over the centrosome where 405-nm light was applied. A significant population of 573-nm REs formed the second population of REs adjacent to the bridge (Fig 2B and C), suggesting that the second population of REs originated from the centrosome

population of REs. Zebrafish pre-abscising cells showed a similar localization of Rab11-REs at the centrosome (Fig 2D) and this organization was similar to what was reported for another RE associated GTPase, Rab25 (Willoughby et al, 2021). These findings suggest REs organization at the centrosome is a conserved process during pre-abscission.

### The oldest mitotic centrosome in pre-abscission moves towards the cytokinetic bridge first and has a more dynamic population of Rab11

One centrosome is inherently older than the other between the two daughter cells because of the nature of centriole duplication. We tested whether the oldest centrosome was more likely to be the predominantly motile centrosome between the two daughter cells. The oldest centrosome can be identified by having elevated centrin-GFP levels (Kuo et al, 2011; Hung et al, 2016; Colicino et al, 2019) (Fig 3A and B). Once the oldest centrosome was noted, live cell imaging was used to test whether the oldest moved towards the bridge before the youngest. Both human and zebrafish pre-abscising cells preferred the oldest centrosome to move towards the cytokinetic bridge over the youngest (Fig 3A–C, 61.5% in zebrafish compared with 80% in HeLa cells).

We next determined if an asymmetry in Rab11 dynamics and organization between the oldest and youngest centrosomes occurred in a pre-abscising cell. Using FRAP, the oldest and youngest centrosomes were identified using centrin-GFP and a ROI was placed over them. mCherry-Rab11a (mCh-Rab11) was photobleached within these regions and recovery was recorded over a 40-s time frame (Fig 3D and E). Mobility of mCh-Rab11 was calculated and depicted as fitted FRAP curves at the oldest centrosome compared with the youngest from three individual pre-abscising cells (Fig 3E). Although there is variation between curves across individual cells (Fig 3E), we find that the oldest centrosome always has an elevated mobile fraction when compared with the younger centrosome in all three pre-abscising cells (Fig 3E). To highlight the difference in mCh-Rab11 dynamics between the oldest and youngest centrosomes between the two daughter cells we took a ratio of the mobile fraction and T1/2 of the oldest/youngest centrosome from the three cells shown in Fig 3E. If there is no difference in the mobile fraction or T1/2, then the ratio should be at one. For the mobile fraction we find a mean of 1.8 ± 0.2 that significantly deviates from one, and for the T1/2 we find a mean ratio of 1.2 ± 0.2 that does not significantly deviate from one (Fig 3F). These studies suggest that mCh-Rab11 has elevated mobility at the oldest centrosome compared with the youngest centrosome.

To examine Rab11 organization between the two mitotic centrosomes during pre-abscission, expansion microscopy was used. Expansion microscopy is a method that improves the final image resolution up to fourfold by physically enlarging structures using a polymer system (Asano et al, 2018; Sahabandu et al, 2019). We expanded our centrin-GFP cells and immunostained them for endogenous Rab11 and GFP (Fig 3G). The oldest and youngest centrosomes were identified by centrin signal enrichment. Strikingly, at the oldest centrosome, Rab11 accumulated around the mother centriole (Fig 3g'). The mother centriole is noted by having elevated centrin levels between the two centrioles. This was

somewhat the case for the youngest mother centriole; however, not to the same extent as the oldest. In both centrosomes, the daughter centrioles significantly lacked Rab11 signal compared with the mother centriole. We quantified the area of Rab11 endosomes that overlapped/touched the mother centriole compared with the daughter centriole (shown in example Fig 3H). We observed that there is significantly higher Rab11 endosome area at the mother centriole (2.66 ± 0.56 $\mu m^2$) compared with the daughter centriole (0.18 ± 0.13 $\mu m^2$, Fig 3I). These studies suggest an asymmetry in Rab11 organization and dynamics between the oldest and youngest centrosome that may contribute to the movement of the centrosome towards the cytokinetic bridge.

### Rab11 GTPase function is associated with centrosome bridge-directed movement

To specifically test the role of Rab11 in centrosome movement, we used a strategy to measure centrosome reorientation during pre-abscission by normalizing distance traveled by the centrosome in relation to the cell (Fig S3A). This involved recording the movement of the centrosome and the cell body as vectors and using vector subtraction to calculate centrosome movement within the reference frame of the cell in both cell culture (Fig S3B) and in zebrafish embryo cells (Fig S3C). We used this strategy with control and Rab11-null, centrin-GFP, or GFP-FIP3 HeLa cell lines (Fig S3D). The Rab11-null cell lines were created using Rab11a CRISPR vector and Rab11a HDR Vector (see supplementary key resource table in Table S1) and characterization done in Rathbun et al (2020b). Western blot analysis to confirm Rab11a loss used an antibody that detects both Rab11a and Rab11b (Fig S3D, supplementary key resource table in Table S1), where a loss of Rab11 signal in our Rab11-null cells compared with control suggests that both Rab11a and Rab11b are not present or that Rab11b is present at low and undetectable amounts. Similar to previous reports, cells null for Rab11a resulted in increased binucleated cells indicative of an abscission failure (Wilson et al, 2005; Rathbun et al, 2020b) that was rescued with ectopic expression of mCh-Rab11a (Fig S3E and F).

In control cells expressing GFP-FIP3, endogenous Rab11 colocalizes with its effector protein, FIP3, during pre-abscission. In Rab11-null cells, no Rab11 is detected by immunohistochemistry and GFP-FIP3 is no longer organized in a tight centrosome localized compartment (Fig 4A). Rab11-null cells were impaired in centrosome movement towards the cytokinetic bridge compared to control cells (Fig 4B and C and Video 3). When measuring the overall centrosome mobility during pre-abscission, centrosomes moved a significantly shorter distance in Rab11-null cells (32.19 ± 2.025 $\mu m$) compared with control (46.50 ± 2.011 $\mu m$, Fig S3G). Although there was centrosome movement in the Rab11-null cells, the centrosome movement was random in its directionality. Thus, we then took distance traveled measurements for cytokinetic bridge directed centrosome movement and identified that the centrosome had significant defects in centrosome-directed distance traveled in Rab11-null cells (2.00 ± 0.86 $\mu m$) compared with controls (8.64 ± 1.63 $\mu m$, Fig 4F), suggesting that Rab11 is needed for directed centrosome movement towards the cytokinetic bridge.

To test the requirement for Rab11 GTPase function, fluorescently tagged versions of Rab11a (referred to as Rab11 from here on) were

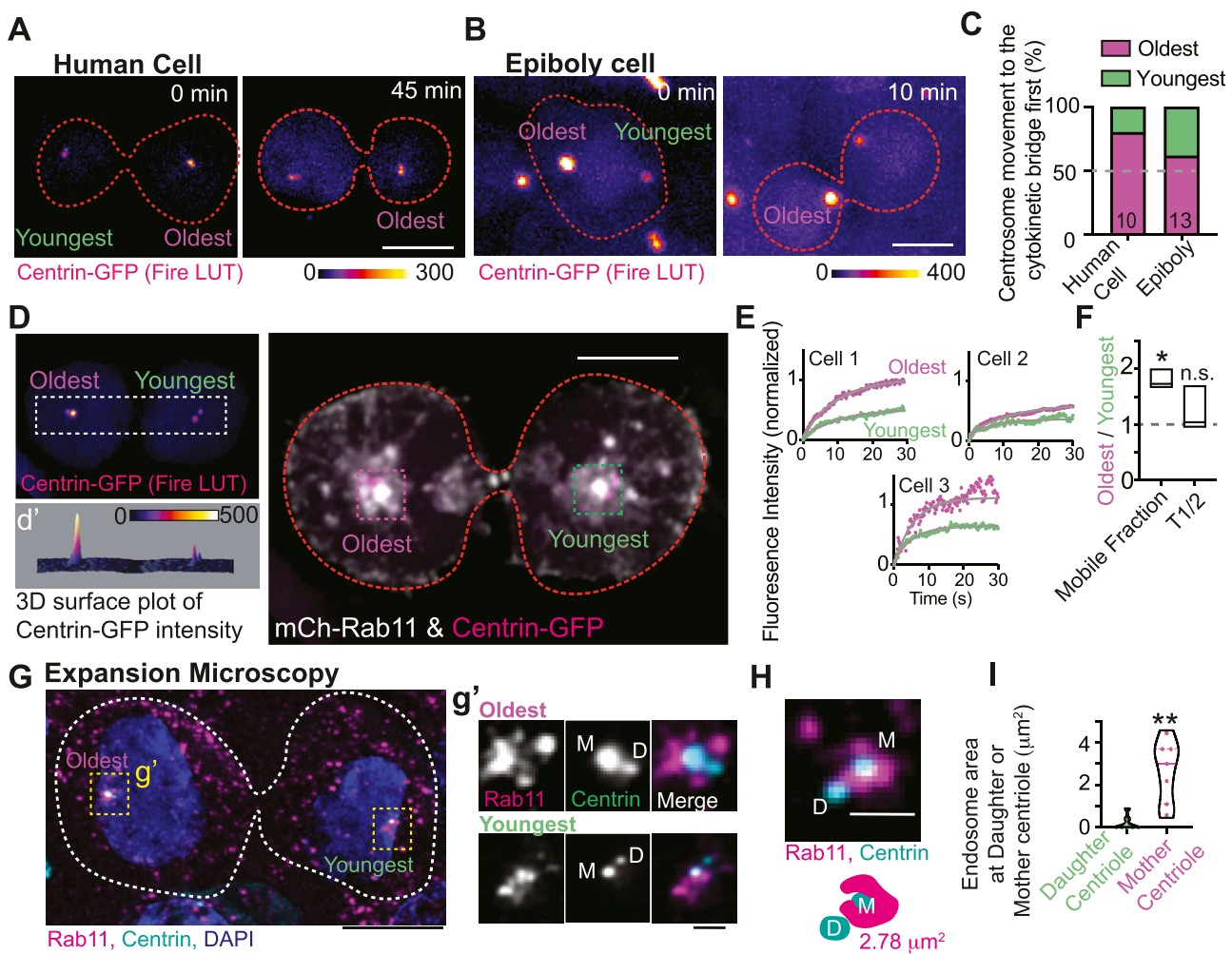

**Figure 3. The oldest mitotic centrosome in pre-abscission moves towards the cytokinetic bridge first and has a more dynamic population of Rab11.**
**(A, B)** Time lapse of a pre-abscising human (HeLa) cell (A) and zebrafish embryo cell at epiboly (B), expressing centrin-GFP (fire LUT). Orange dashed lines, cell boundaries. Scale bar, 10 μm. **(C)** Percentage of daughter cells with centrosome movement towards the cytokinetic bridge in zebrafish embryos and human (HeLa) cells. n-values on graph, Table S1. **(D)** Pre-abscising human (HeLa) cell expressing centrin-GFP (fire LUT, magenta in merge) and mCh-Rab11 (gray). **(d')** Three-dimensional surface plot of centrin-GFP to note oldest and youngest centrosome. Orange dashed lines, cell boundaries. Scale bar 10 μm. **(E)** FRAP traces of mCh-Rab11 at the oldest mitotic centrosome (magenta) and youngest mitotic centrosome (green) from three cells depicted. **(F)** mCh-Rab11 mobile fraction and half-life (T1/2) were calculated as a ratio of oldest centrosome/youngest centrosome and presented as a box and whisker plot with mean (dark line). n = 3. Minimum and maximum values noted by the boxed boundaries. n-values and statistical results detailed in Table S1. **(G, H)** Expanded pre-abscising human (HeLa) cell expressing centrin-GFP (cyan) and immunostained with Rab11 (magenta) and DNA (DAPI, blue). Dashed lines, cell boundaries. Scale bar, 25 μm. **(g')** Magnified insets from (G) depicting Rab11 (gray, magenta in merged) and centrin-GFP (gray, cyan in merged). **(H)** Example pre-abscising expanded cell centrosome with a model below demonstrating Rab11-endosome area at the mother centriole. **(g', H)** M, denotes mother centriole and D, denotes daughter centriole. Scale bar, 2.5 μm. **(I)** Endosome area at daughter or mother centriole from pre-abscising cells quantified and presented as a violin plot with median (thick line) and quartiles (dotted lines), n = 7 centrosomes, across n = 4 cells. Two-tailed t test, **P < 0.01. n-values and statistical results detailed in Table S1.

expressed in Rab11-null cells. These included Rab11 (mCh-Rab11), a mutant that mimic's the GDP-bound state of Rab11 (mCh-Rab11(S25N)), or a mutant that mimics the GTP-bound state of Rab11 (mCh-Rab11(Q70L), Fig 4D). Comparable expression levels of ectopically expressed Rab11 demonstrated by Western blot in Fig S3D. mCh-Rab11 and -Rab11(Q70L) had similar centrosome localization patterns whereas -Rab11(S25N) remained more cytosolic (Fig 4D). Expressing mCh-Rab11 in Rab11-null cells rescued bridge-directed centrosome movement with a bridge-directed distance traveled of 4.57 ± 1.46 μm, compared with -Rab11(S25N) (1.93 ± 1.17 μm) or -Rab11(Q70L) (0.92 ± 0.92 μm) (Figs 4C, E, and F and Video 4). These findings suggest that the ability of Rab11 to cycle between a

GTP and GDP bound state is important for its contribution to centrosome movement towards the cytokinetic bridge.

We were surprised that mCh-Rab11(Q70L) did not partially rescue centrosome movement towards the cytokinetic bridge because it can mimic the active state of Rab11 and localizes to centrosomes (Fig 4D). One possibility is that GTP to GDP cycling is required for Rab11 to regulate centrosome bridge-directed movement and affects the temporal and spatial organization of Rab11 at the mother centriole during pre-abscission. To test this, we examined whether a difference in mCh-Rab11 and -Rab11(Q70L) dynamics occurred using FRAP by photobleaching the population of mCh-Rab11 or -Rab11(Q70L) at the centrosome and comparing the fluorescent

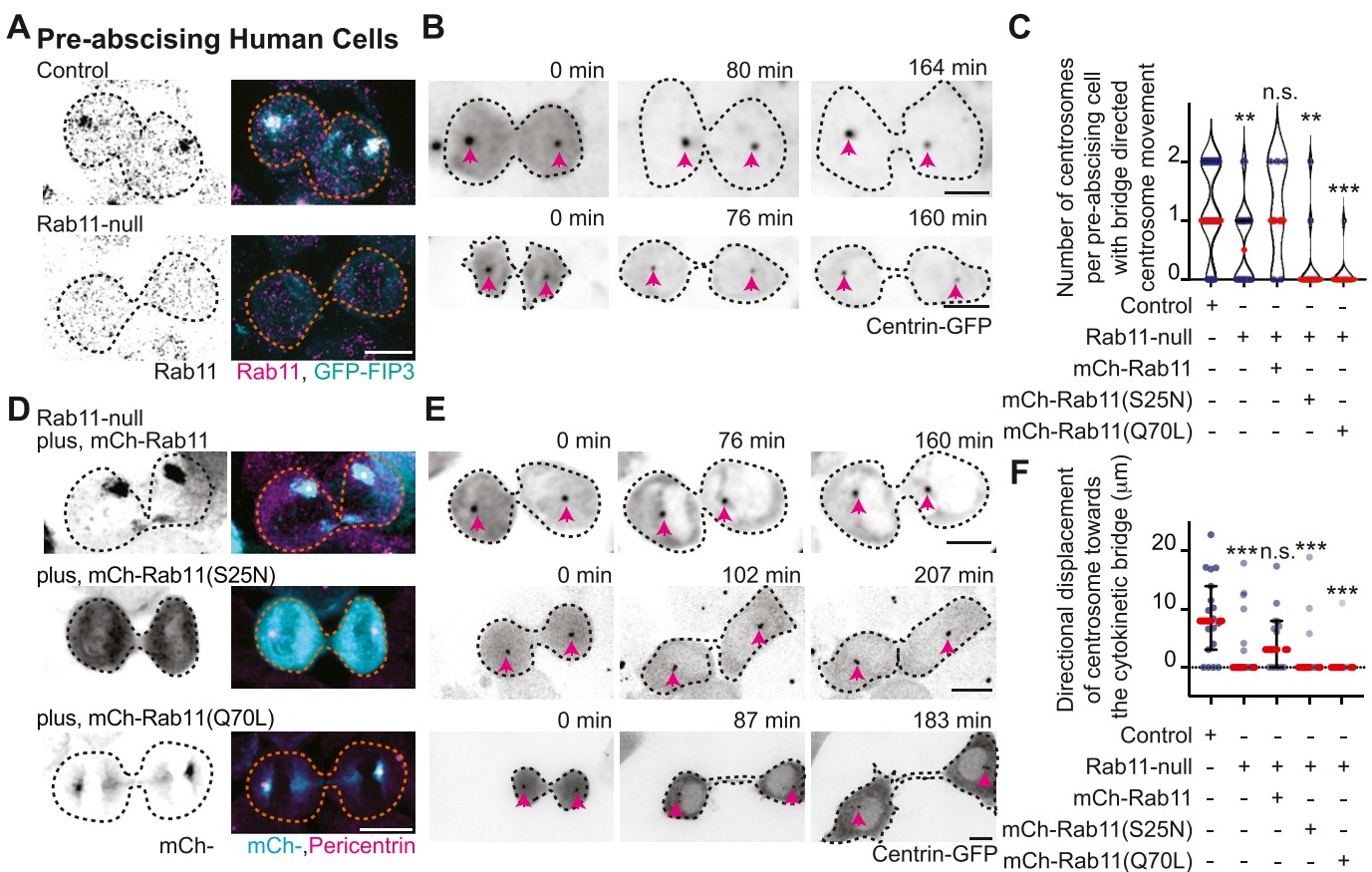

**Figure 4. Rab11 GTPase function is associated with centrosome bridge-directed movement.**
**(A)** Control and Rab11-null expressing GFP-FIP3 (cyan, merge) human (HeLa) cells fixed and immunostained for Rab11 (inverted grays, left; magenta, merge). **(B)** Time-lapse of control and Rab11-null centrin-GFP (inverted grays) pre-abscising human (HeLa) cells. Pink arrows, centrosome. Dashed lines, cell boundaries (A, B). Scale bar, 10 μm Video 3. **(C, D, E, F)** Rab11-null cells expressing mCh-Rab11, -Rab11 (S25N), or -Rab11(Q70L). **(C)** Number of centrosomes per pre-abscising cell with bridge directed centrosome movement calculated as both centrosomes (2 centrosomes), only one centrosome (1 centrosome) and neither centrosome (0 centrosomes) moved shown as a violin plot with median (orange dashed line) and quartiles (dark dotted lines). One-way ANOVA, with Dunnett's multiple comparison to control, ***$P < 0.001$, **$P < 0.01$ and n.s. not significant. **(D)** GFP-FIP3 Rab11-null cells expressing mCh-Rab11, -Rab11(S25N) or -Rab11(Q70L) (inverted grays, left; cyan, merge) fixed and immunostained for Pericentrin (magenta, merge). **(E)** Centrin-GFP (inverted grays) Rab11-null pre-abscising human (HeLa) cells expressing mCh-Rab11, -Rab11(S25N), or -Rab11(Q70L). Pink arrows, centrosome. Dashed lines, cell boundaries (D, E). Scale bar, 10 μm Video 4. **(F)** Directional displacement of centrosome towards cytokinetic bridge (scatter plot). Median (orange dashed line) and quartiles (dark lines) shown. One-way ANOVA with Dunnett's multiple comparison to control, ***$P < 0.001$ and n.s. not significant. **(C, F)** n values, statistical results detailed in Table S1.

recovery over time. We selected cells with a similar range of fluorescent intensities of mCh-Rab11 and -Rab11(Q70L) (Fig S3I) for these studies and found significant differences in dynamics between mCh-Rab11 and -Rab11(Q70L) (Fig S3H–L). Rab11(Q70L) cells had a decreased mobile fraction of 36.57% ± 4.48%, compared with Rab11 at 59.02% ± 5.08% and Rab11(Q70L) presented with a decreased half-life (2.27 ± 0.33 s) compared with Rab11 (9.00 ± 1.31 s, Fig S3I–L). One potential interpretation is that Rab11(Q70L) is more stably associated with the mother centriole (as seen in Fig 3g'). Collectively, these findings suggest that mCh-Rab11 dynamics at centrosomes is partly regulated by its ability to cycle GTP to GDP contributing to centrosome movement towards the cytokinetic bridge.

### Rab11 GTP-cycling mediates centrosome protein, Pericentrin, centrosome localization during pre-abscission

We find that γ-tubulin and Pericentrin partially localize to REs during pre-abscission in both human (HeLa) and zebrafish embryo

cells (Fig S4A and B), which is consistent with previous studies that identified Pericentrin and γ-tubulin at REs during metaphase in human cells (Hehnly & Doxsey, 2014). We specifically find that REs associate with Pericentrin and γ-tubulin at and outside of the centrosome (Fig S4A and B, inset demonstrating acentrosomal localization). In human (HeLa) cells expressing the RE marker, GFP-FIP3 and the centrosome-targeting domain from Pericentrin, DsRed-PACT, and GFP-FIP3 colocalized with acentrosomal DsRed-PACT. These GFP-FIP3/DsRed-PACT puncta move towards and combine with the main centrosome (Fig S4C) suggesting that REs may help contribute to overall centrosome organization and function.

Because Rab11-null pre-abscising cells are unable to orient their centrosome towards the cytokinetic bridge, we analyzed whether localization of centrosome components and RE components at the centrosome were altered in Rab11-null cells compared with control cells (Figs 5A–D and S4D and E). Fluorescence intensity was measured in fixed early-abscission cells where the centrosomes are still

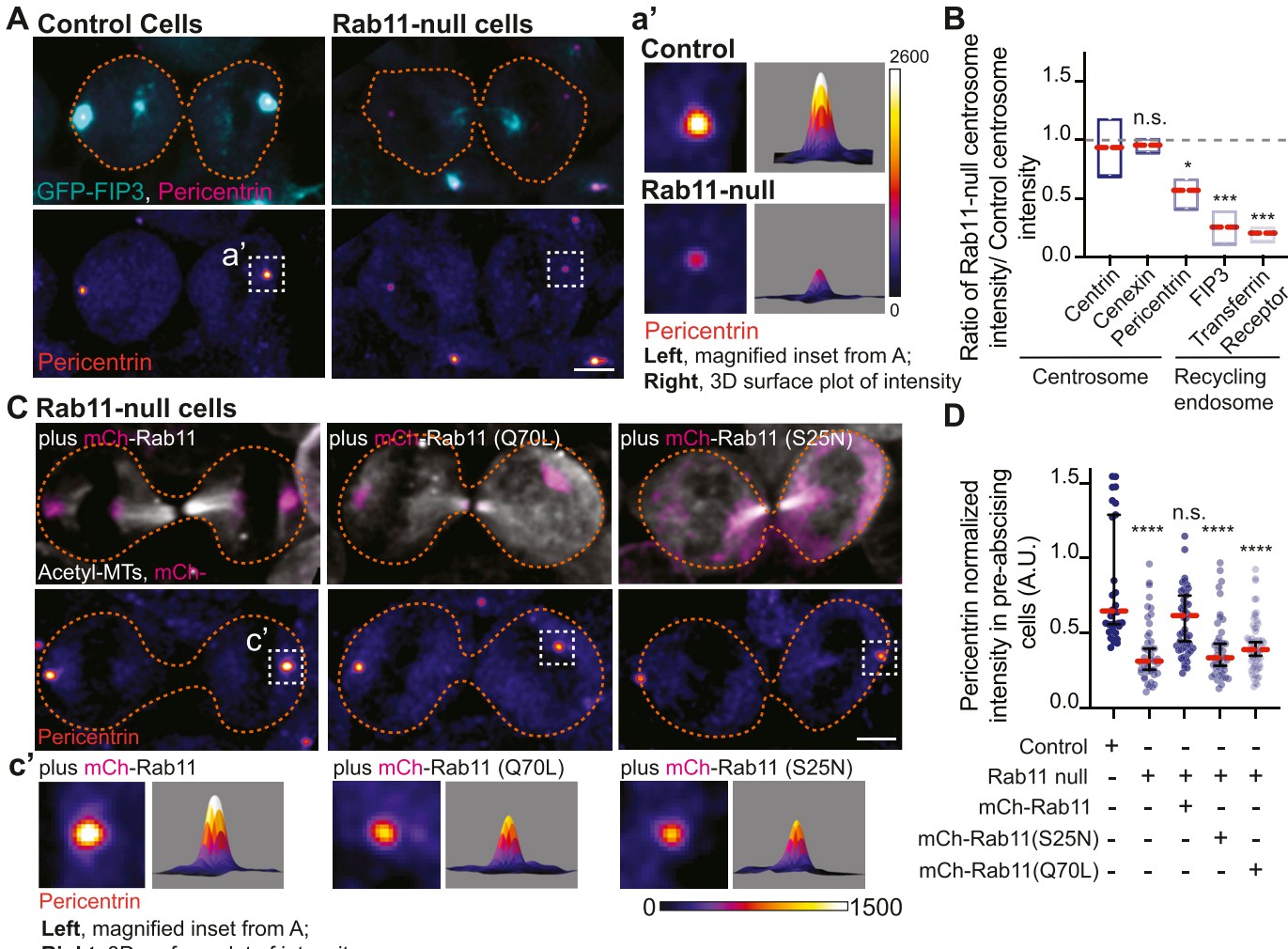

**Figure 5. Rab11 GTP cycling mediates centrosome protein, Pericentrin, centrosome localization during pre-abscission.**
**(A)** Human (HeLa) pre-abscising cells expressing GFP-FIP3 (cyan) were fixed and immunostained for Pericentrin (magenta, top panel; fire LUT, bottom panel). Magnified insets (3×) shown on right (a') with associated three-dimensional surface plot of intensity. Scale bar, 5 μm. **(B)** Box and whisker plot with mean (orange dashed line) depicting ratio of Rab11-null centrosome intensity over control centrosome intensity of cenexin, centrin, Pericentrin, FIP3, and transferrin receptor. Minimum and maximum values noted by boxed boundaries. n > 30 centrosomes per experiment across n = 3 experiments. One-way ANOVA with Dunnett's multiple comparison to centrin, n.s. not significant, *$P < 0.05$ and ***$P < 0.001$. **(C)** GFP-FIP3 Rab11-null pre-abscising human (HeLa) cells ectopically expressing mCh-Rab11, -Rab11(S25N) or -Rab11(Q70L) (magenta, top panel) were fixed and immunolabeled for acetylated-tubulin (gray, top panel) and Pericentrin (fire LUT, bottom panel). Scale bar, 5 μm. **(c')** 3× magnified centrosome inset (left), 3D fluorescent intensity surface plot of inset (right). **(D)** Scatter plot with median (orange dashed line) and quartiles (dark lines) depicting normalized Pericentrin intensities at centrosomes from pre-abscising human (HeLa) cells. One-way ANOVA with Dunnett's multiple comparison to control, n.s. not significant and ****$P < 0.0001$. n > 38 centrosomes across n > 3 experiments. **(B, D)** n values and statistical results detailed in Table S1.

on the polar ends of the daughter cells before they move to the bridge. An ROI was placed over the centrosome and compared between control and Rab11-null cells. The fluorescence intensity values were plotted as a ratio of Rab11-null over control cells (Fig 5B). A ratio of 1 implies no difference between Rab11-null cells and control, and a ratio significantly less than 1 suggests decreased centrosome localization in Rab11-null cells. Rab11-null cells had significantly decreased centrosome localized GFP-FIP3 (Figs 5A and B and S4D), transferrin receptor (RE cargo, Figs 5B and S4D), and Pericentrin (Fig 5A and B). However, the centriole appendage protein, cenexin, and centriole protein, centrin, were not affected by Rab11 loss (Figs 5B and S4E). This is interesting because we note Rab11 endosomes at the mother centriole (Fig 3G–I) suggesting that

Rab11 may interact with mother centriole appendages (as shown for interphase cells, [Hehnly et al, 2012]), but does not seem to affect their structure during pre-abscission (Figs 5B and S4E). However, Rab11-REs do seem to be implicated in Pericentrin organization both in early stages of pre-abscission (Fig 5A and B) and later stages (Fig S4F and G). Pericentrin intensity at the centrosome was rescued in Rab11-null cells by expression of mCh-Rab11, but not mCh-Rab11(S25N) or -Rab11(Q70L) (Fig 5C and D). Pericentrin expression levels were unaltered in Rab11-null cells, or cells rescued with mCh-Rab11, -Rab11(S25N), or -Rab11(Q70L) (Fig S3D), suggesting that the targeted localization of Pericentrin is disrupted with loss of Rab11 function. Taken together, Rab11 GTPase function is necessary for centrosome movement towards the bridge during

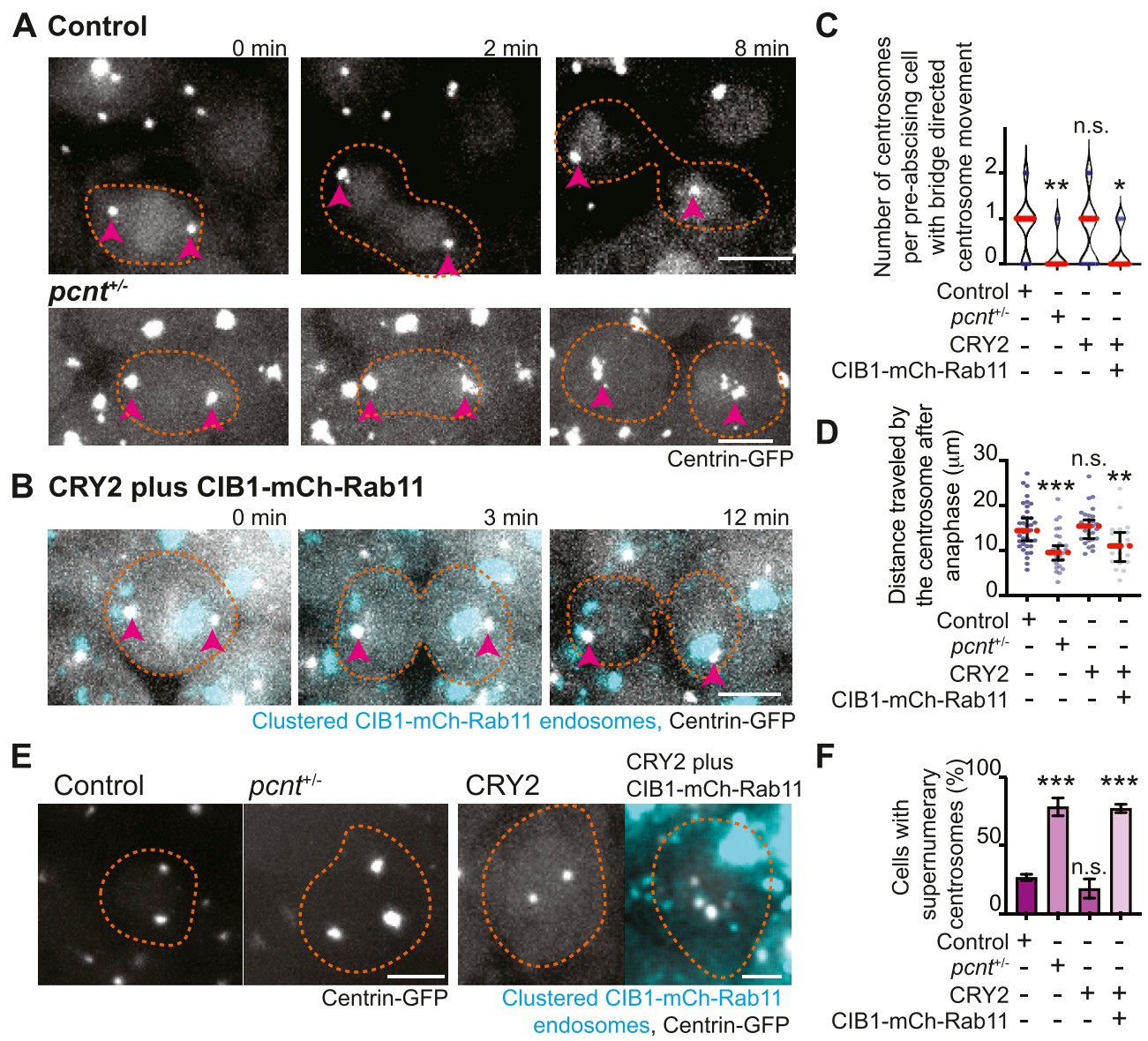

**Figure 6. Pericentrin and Rab11 endosomes coordinate centrosome movement and number during mitotic exit.**
**(A, B)** Time-lapse imaging of control cells (A), *pcnt*$^{+/-}$ cells (A), and cells with clustered Rab11 endosomes (B, cyan) from a -5actb2:cent4-GFP(centrin-GFP, gray) embryo. Pink arrows, centrosome. Dashed orange line, cell boundaries. Scale bar, 10 *μm*. **(C, D)** Number of centrosomes per pre-abscising cell with bridge directed centrosome movement calculated as both centrosomes (2 centrosomes), only one centrosome (1 centrosome) and neither centrosome (0 centrosomes) moved depicted as a violin plot with median (orange dashed line) and quartiles (dark dotted lines, C), and distance traveled by the centrosome after anaphase (scatter plot with median, orange dashed line, and quartiles, dark lines, D) in control, *pcnt*$^{+/-}$, CRY2 injected, or CRY2 plus CIB1-mCh-Rab11 injected embryos is shown. n > 16 cells across n > 3 embryos. One-way ANOVA with Dunnett's multiple comparison to control, n.s. not significant, *$P < 0.05$, **$P < 0.01$ and ***$P < 0.001$. **(E, F)** Interphase centrin-GFP (gray) zebrafish embryos with n = 2 centrosomes or supernumerary (n > 2 centrosomes) in control, *pcnt*$^{+/-}$, CRY2 injected, CRY2 plus CIB1-Rab11 injected embryos (cyan, CIB1-mCh-Rab11 plus CRY2). **(E)** Orange dashed lines, cell boundaries. Scale bar, 5 *μm*. **(F)** Percentage of cells with supernumerary centrosomes (n > 2 centrosomes). n > 30 cells per embryo across n > 3 embryos. One-way ANOVA with Dunnett's multiple comparison to control, n.s. not significant and ***$P < 0.001$. **(C, D, F)** n values and statistical results detailed in Table S1.

pre-abscission (Fig 4) by potentially facilitating Pericentrin organization at the centrosome (Fig 5).

## Pericentrin and Rab11 endosomes coordinate centrosome movement and number during mitotic exit

Because loss of Rab11 caused a decrease in Pericentrin levels at the centrosome in human cells (Fig 5), we tested the role of Pericentrin and Rab11-endosomes in regulating centrosome movement during pre-abscission in vivo. To do this, we used heterozygous Pericentrin (*pcnt*$^{+/-}$) zebrafish embryos (Sepulveda et al, 2018) positive for the centrosome marker centrin-GFP (embryo generation modeled in Fig S5A) or by acutely inhibiting Rab11-associated vesicles through an optogenetic oligomerization approach that relies on a hetero-interaction between CRY2 and CIB1 that is induced by the application of blue light (modeled in Fig S5B, [Rathbun et al, 2020b]).

$Pcnt^{+/-}$ embryos were used because of our inability to obtain reliable $pcnt$ null embryos. When comparing $pcnt^{+/-}$ and Rab11 optogenetically clustered embryos with control embryos (non-injected or CRY2 injected) at 3.3–5 hpf, $pcnt^{+/-}$ embryo cells and Rab11 clustered embryo cells were significantly impaired at reorienting their centrosomes towards the cytokinetic bridge compared with control conditions (Fig 6A–C, confirmation of genotype in Fig S5C, CRY2 control in Fig S5D). $Pcnt^{+/-}$ and Rab11 clustered embryos also demonstrated significant defects in centrosome movement during pre-abscission, where the calculated distance traveled was significantly decreased for $pcnt^{+/-}$ and Rab11 clustered embryos compared with control conditions (Fig 6D). These studies suggest that both Pericentrin and Rab11 coordinate centrosome bridge directed motility.

We next examined whether decreasing Pericentrin ($pcnt^{+/-}$) and acutely clustering Rab11-vesicles resulted in abscission defects. When cells are unable to complete abscission, cells become binucleated (Carter, 1967) or present with supernumerary centrosomes (Pihan et al, 2003). Most cells contain either one or two centrosomes, depending on their cell cycle stage. A G1 cell contains a single centrosome composed of two centrioles. During the S phase, two new (daughter) centrioles assemble near the pre-existing (mother) centriole creating two centrosomes that will move apart to create a bipolar mitotic spindle. When cells fail abscission, the two daughter cells combine gaining an extra centrosome and can present as binucleated. Having too many centrosomes can result in multipolar divisions, chromosome segregation defects, defects in asymmetric cell division, loss in cell polarity, induction of invasive protrusions, and inappropriate activation of signaling pathways (Godinho & Pellman, 2014). In our studies, we counted embryonic cells that had an abnormal number of centrosomes (three or more) or were binucleated. With $pcnt^{+/-}$ embryos, we find a significant increase in embryos with binucleated cells (Fig S5E and G) and supernumerary centrosomes (78.33% ± 6.509% of cells) compared with control (26.57% ± 2.293%, Fig 6E and F). With optogenetic clustering of Rab11-REs, we find a similar dramatic increase in both the percentage of cells containing supernumerary centrosomes (77.18% ± 3.029%) compared with CRY2-injected controls (18.41% ± 7.079%, Fig 6E and F) and binucleated cells (Fig S5F and G), suggesting that Rab11-endosomes and Pericentrin may work together to coordinate successful abscission.

## Discussion

We propose a model where the cytokinetic bridge and associated midbody may act as a symmetry breaking event marking a position for where the centrosome needs to reorient towards (Fig 1). This reorientation is particularly interesting in the context of KV development, where we find in these studies (Fig 1B–D) and in our previous studies (Rathbun et al, 2020b) that during pre-abscission, the two daughter cells position themselves so that the cytokinetic bridge is placed where a lumen will form. Herein we found that centrosomes reorient toward the cytokinetic bridge before bridge cleavage. Although this is not specific to just KV cells that are destined to assemble a primary cilium at this locale, providing a

biologically useful role for centrosome reorientation to this cite, it also occurs in cells dividing during epiboly. This suggests that centrosome reorientation during pre-abscission may have two underlying functions: (1) centrosome reorientation needs to occur to make sure the cilia forms in the correct place when destined ciliated cells need to self-assemble into a rosette structure that transitions to form a lumen de novo, (2) that centrosome reorientation towards the bridge may play a role in successful abscission completion. Our studies that support this with pre-abscising zebrafish epiboly cells containing optogenetically clustered Rab11 membranes or $pcnt^{+/-}$ cells that present with significant defects in reorienting their centrosomes towards the cytokinetic bridge. Associated with these defects is a significantly greater propensity for these cells to be binucleated and contain supernumerary centrosomes suggestive of an abscission defect. These studies suggest that centrosome function, Rab11-endosome function, and centrosome reorientation may be needed for abscission completion.

Rab11 has previously been identified to interact with sub-distal appendages in non-ciliated interphase cells (Hehnly et al, 2012). An additional report demonstrated that Rab8, and potentially the Rab11-Rab8 cascade, has an association with distal appendages during ciliogenesis (Schmidt et al, 2012) suggesting that at different cell cycle stages different flavors of Rab11 or RE interactions may occur with the centrosome. It has yet to be reported if there is a relationship between Rab11-endosomes and the mother centriole during mitotic stages. Our findings demonstrate using expansion microscopy that Rab11 continues to organize at the mother centriole during pre-abscission (Fig 3G–I) and it will be exciting to investigate whether this organization is dependent on sub-distal or distal appendages. There are ample reports that Rab11 is involved in abscission (Wilson et al, 2005; Schiel et al, 2012), but the relationship of Rab11 with the centrosome during this time had yet to be investigated. Thus, our study is confirming and expanding on the initial foundational studies where we suggest an importance for Rab11-endosome interaction with the centrosome during multiple cell cycle stages. Specifically, we find that Rab11 is needed for centrosome-directed movement towards the cytokinetic bridge.

Connections between the centrosome and cleavage of the cytokinetic bridge exist. For instance, the centrosome protein, Cep55, localizes both to the centrosome and cytokinetic midbody where it contributes to abscission (Little & Dwyer, 2021). Piel et al (2001)'s work was foundational in identifying the centrosome as a possible regulator of abscission where they performed centrosome-removal experiments (Piel et al, 2001). Centrosome-free prostate epithelial cells (Wang et al, 2020), human embryonic stem cells (Renzova et al, 2018), or BSC1 cells (Piel et al, 2001) obtained by either centrinone treatment or by physically severing a portion of the cell containing the centrosome from the cellular nucleus (Piel et al, 2001) present with abscission defects such as multi-nucleation. Severing the centrosome from the cell could potentially remove Rab11-associated REs affiliated with the centrosome (Fig 2). Rab11-REs are needed for abscission (Figs S3E and F, 6E and F, and S5F and G) (Wilson et al, 2005). Here we identified a connection with Rab11 and centrosome function, such that when Rab11 is removed centrosome levels of Pericentrin significantly decrease (Fig 5A and B). A loss in Pericentrin levels then leads to centrosome reorientation defects

and abscission failure (Fig 6A, C, D, and F). This suggests a potential molecular mechanism where Rab11 modulates Pericentrin centrosome organization that is needed in centrosome re-orientation to the cytokinetic bridge and likely in its subsequent abscission. Taken together, these previous studies and our studies herein suggest that the centrosome has a role in the process of abscission.

# Materials and Methods

## Resource availability

### Lead contact
For further information or to request resources/reagents, contact the Lead Contact, Heidi Hehnly (hhehnly@syr.edu).

### Materials availability
New materials generated for this study are available for distribution.

### Data and code availability
All data sets analyzed for this study are displayed.

## Experimental model and subject details

### Fish lines
Zebrafish lines were maintained using standard procedures approved by Syracuse University IACUC (Institutional Animal Care Committee) (Protocol #18-006). Embryos were raised at 28.5°C and staged (as described in reference Kimmel et al [1995]). Control and/or transgenic zebrafish lines used for live imaging and immuno-histochemistry are listed in supplementary key resource table in Table S1.

## Method details

### Antibodies
Antibody catalog information used in HeLa cells and zebrafish embryos are detailed in supplementary key resource table in Table S1.

### Plasmids and mRNA
Plasmids were generated using Gibson Cloning methods (NEBuilder HiFi DNA assembly Cloning Kit) and maxi-prepped before injection and/or transfection. mRNA was made using mMESSAGE mMACHINE SP6 transcription kit. See supplementary key resource table in Table S1 for a list of plasmid constructs and mRNA used.

### Cell culture
HeLa cells stably expressing GFP-FIP3 or centrin-GFP (from Piel et al [2001], Wilson et al [2005], Kuo et al [2011], Hehnly et al [2012], and Hehnly and Doxsey [2014]) were maintained at 37°C with 5% $CO_2$. Rab11 CRISPR vector and Rab11a HDR vector were transfected into cells using Mirus TransIT-LT1 transfection reagent (supplementary key resource table in Table S1). Cells were selected in puromycin

(5 µg/ml). Rab11a-null cells were transfected with mCh-Rab11a (pCS2-mCh-Rab11a), dominant negative Rab11a (pCS2-mCh-Rab11(S25N)) and constitutively active Rab11 (pCS2-mCh-Rab11(Q70L)) using Mirus TransIT-LT1. Cells were tested for Rab11 levels using a Western blot with an antibody that detects both Rab11a and Rab11b.

### Western blot
Cell lysates were acquired by suspending cells in lysis buffer (HSEG buffer, pH 7.4, 40 mM HEPES, 40 mM NaCl, 5 mM EDTA, 4% glycerol, 20 mM NaF; 1% Triton X-100; 1X protease inhibitor; and 0.1 mM PMSF). After collecting post-nuclear supernatant from lysates, protein concentration was calculated using Bio-Rad Protein Assay Kit II (see supplementary key resource table in Table S1). Standard Western blot procedures were performed. Nitrocellulose membranes were probed with primary antibody and/or primary antibody conjugated to horseradish peroxidase diluted in TBS-Tween20 and incubated overnight at 4°C. The membranes were probed using appropriate secondary antibody for an hour at room temperature. The protein levels were visualized using Clarity Western ECL substrate (see supplementary key resource table in Table S1) and imaged using Bio-Rad ChemiDoc imager.

### Immunofluorescence
Cells were plated on #1.5 coverslips until they reach 90% confluence fixed in 4% PFA at room temperature (30 min) or 100% ice cold methanol (10 min). Standard immunofluorescent procedures were performed for PFA fixation (Hehnly et al, 2006) and for methanol (Colicino et al, 2018). Coverslips were rinsed with $dH_2O$ and mounted on glass slides using either Prolong Diamond with DAPI mounting media or Prolong Gold (see supplementary key resource table in Table S1). For zebrafish embryo immunofluorescent protocols, see Aljiboury et al (2021).

### Expansion microscopy
Cells were plated on #1.5 coverslips until they reach 90% confluence and fixed with 4% PFA at room temperature. Standard immuno-fluorescent procedures were performed for PFA fixation (Hehnly et al, 2006). Expansion microscopy was modified from (Asano et al, 2018; Sahabandu et al, 2019). Specifically, 20% acrylamide gels (gelation solution 20% acrylamide, 7% sodium acrylate, 0.04% bis acrylamide, 0.5% APS, and 0.5% TEMED in 1× PBS) were poured on coverslips and allowed to solidify on ice. The solidified gels were then carefully sectioned into 4 mm gel punches using a disposable biopsy punch on ice. These gel punches were then digested with digestion buffer overnight (0.5% Triton-X, 0.03% EDTA, 1M Tris–HCl, pH 8, and 11.7% sodium chloride). We did not use proteinase K in the digestion step to avoid disruption of centrosome proteins. The gel punches were then subjected to a second round of immunofluo-rescence procedures with antibody concentration titrated down to half the initial concentration. They were then expanded in water at room temperature for 2 h with water exchanged every 20 min. The expanded punches were then mounted on to MatTek plates and imaged on the Leica SP8 confocal microscope with Lightning and using a long-range objective (HC PL APO 40×/1.10 W CORR CS2 0.65 water objective) to be able to view the sample effectively through the thickness of the agar.

Rab11-endosome area was quantified at the oldest and youngest centriole (differentiated by elevated levels of centrin-GFP signal at the oldest centriole, [Kuo et al, 2011]). The area around the centriole was quantified by drawing an ROI around Rab11-endosomes that overlapped/touched the centrioles. The Rab11-endosome areas were then compared between the oldest and youngest centriole using a violin plot, graphed using the PRISM9 software.

### Genotyping pcnt^{+/−} zebrafish

Tail fins of adult zebrafish were clipped, whole embryos or fixed and stained embryos were used to extract genomic DNA and genotyped according to Sepulveda et al (2018).

### Imaging

Zebrafish and tissue culture cells were imaged using Leica DMi8 (Leica) equipped with a X-light V2 Confocal unit spinning disk equipped with a Visitron VisiFRAP-DC photokinetics unit, a Leica SP8 (Leica) laser scanning confocal microscope (LSCM) and/or a Zeiss LSM 980 (Carl Zeiss) with Airyscan 2 confocal microscope. The Leica DMi8 is equipped with a Lumencore SPECTRA X (Lumencore), Photometrics Prime-95B sCMOS Camera, and 89 North-LDI laser launch. VisiView software was used to acquire images. Optics used with this unit are HC PL APO ×40/1.10W CORR CS2 0.65 water immersion objective, HC PL APO ×40/0.95 NA CORR dry and an HCX PL APO ×63/1.40-0.06 NA oil objective. The SP8 laser scanning confocal microscope is equipped with HC PL APO 20×/0.75 IMM CORR CS2 objective, HC PL APO 40×/1.10 W CORR CS2 0.65 water objective and HC PL APO ×63/1.3 Glyc CORR CS2 glycerol objective. LAS-X software. was used to acquire images. The Zeiss LSM 980 is equipped with a T-PMT, GaAsP detector, MA-PMT, Airyscan 2 multiplex with 4Y and 8Y. Optics used with this unit are PL APO 63×/1.4 NA oil DIC. Zeiss Zen 3.2 was used to acquire the images. A Leica M165 FC stereomicroscope equipped with DFC 9000 GT sCMOS camera was used for staging and phenotypic analysis of zebrafish embryos.

For live human cell imaging, cells were plated on #1.5 glass bottom Ibidi slides or MatTek plates (see supplementary key resource table in Table S1). Cells were imaged using spinning disk confocal, LSCM, or wide-field fluorescent microscopy followed by deconvolution (AutoQuant X3). Cells were imaged in a temperature- and $CO_2$-controlled chamber for 1–10 h at 0.5 to 4-min time intervals.

For zebrafish embryo imaging, fluorescent transgenic or mRNA injected embryos (refer to strains and mRNAs in supplementary key resource table in Table S1, and for injection protocols refer to Aljiboury et al [2021]) were embedded in 2% agarose at 3.3–5 hpf and imaged using the spinning disk or LSCM.

### FRAP and photoconversion

FRAP experiments to compare mobility of mCh-Rab11 and -Rab11 (Q70L) at the centrosome were conducted 24 h post transfection of mCh-Rab11 or -Rab11(Q70L) in centrin-GFP cells using the Leica DMi8 with spinning disk and photokinetics unit (Visitron VisiFRAP-DC). A ROI was marked at the centrosome in a cyto-kinetic cell and a 405-nm laser was used to photobleach mCherry within that region. After photobleaching, the cell was imaged live to identify recovery of fluorescent signal at the centrosome at 3 s intervals for 3 min.

FRAP experiments to compare whether mobility of mCh-Rab11 at the centrosome was dependent on the age of the centrosome were conducted 24 h post transfection of mCh-Rab11 in centrin-GFP cells using the Leica DMi8 with spinning disk and photokinetics unit (Visitron VisiFRAP-DC). The age of the centrosome was determined by using elevated centrin-GFP levels at the centrosome to identify the oldest centrosome (Kuo et al, 2011). A ROI was marked at the centrosomes in a cytokinetic cells, and a 405 nm laser was used to photobleach mCh within that region. After photobleaching, the cell was imaged live to identify recovery of fluorescent signal at the centrosome for 40 s. The ImageJ FRAP calculator macro plug-in was used to generate FRAP curves and calculate half-life and immobile fraction values. Graphs were generated using PRISM9 software.

For photoconversion experiments Dendra-Rab11 was expressed in HeLa cells. A ROI was placed over centrosome localized Dendra-Rab11 in a single daughter cell during pre-abscission. A 405 nm laser was applied within the ROI to photoconvert Dendra-Rab11 from green emission (507 nm) to a red emission (573 nm).

### Rab11 optogenetics experiments in zebrafish

Tg (-5actb2:cent4-GFP), Tg (sox17:GFP-CAAX) (Dasgupta & Amack, 2016), and Tg BAC(cftr-GFP) (Navis et al, 2013) zebrafish embryos were injected with 50–100 pg of CIB1-mCh-Rab11, CIB1-mCerulean-Rab11, CRY2-mCherry, and/or CRY2 mRNA at the one cell to four-cell stage (Rathbun et al, 2020a). Embryos were allowed to develop to a minimum of 3.5 hpf and exposed to 488 nm light, while being imaged using the spinning disk confocal microscope.

### Number of centrosomes per pre-abscising cell with bridge directed centrosome movement

Human (HeLa) cells expressing centrin-GFP or DsRed-PACT and zebrafish embryos expressing centrin-GFP and/or PLK1-mCh were imaged using wide-field or confocal based imaging. Cells were monitored for centrosome movement using FIJI/ImageJ. The number of centrosomes that moved towards the cytokinetic bridge was quantified as both daughter cell centrosomes (2 centrosomes), only one daughter cell centrosome (1 centrosome) and neither centrosome (1 centrosomes). Centrosome movement was defined as movement from the polar end of the daughter cell across the centroid of the cell towards the cytokinetic bridge (green circle, modeled in Fig S1D). Findings were plotted as a violin plot using PRISM9.

### Distance of the centrosome from cytokinetic bridge

Cells stably expressing centrin-GFP were live imaged after ana-phase exit and were monitored for centrosome movement. Centrosome movement towards the cytokinetic bridge was quantified based on centrosome movement across the centroid of the cell towards the cytokinetic bridge (green circle, Fig S1C modeled in Fig S1D). Distance of the centrosome from the cytokinetic bridge was quantified at the time point the centrosome was nearest to the bridge by using the line tool on FIJI to trace a line to the midpoint of the cytokinetic bridge (modeled in Fig S1D). Frequency distribution of the distance of the centrosome from the cytokinetic bridge was calculated and graphed as a histogram denoting relative frequency (percentage of pre-abscising cells) that moves the centrosome to at

least 2, 4, 6, and 8 $\mu m$ from the midway point of the cytokinetic bridge using PRISM9.

### Tracking centrosome movement

Human (HeLa) cells and zebrafish embryos expressing centrin-GFP were imaged using widefield or confocal based imaging. Cells were projected and the manual tracking plug-in (FIJI/ImageJ) was used to track the movement of the centrosome (centrin-GFP) from metaphase exit until 3 h post anaphase in HeLa cells and up to 12 min post anaphase or abscission completion in zebrafish embryos. The X and Y coordinates of the centrosome and cell body were recorded at each time-point. The change in position of X and Y of the cell body, marked as the center of the cell at each time point ($\Delta X_{Cellbody} = X_{cellbody-t2} - X_{Cellbody-t1}$; $\Delta Y_{Cellbody} = Y_{Cellbody-t2} - Y_{Cellbody-t1}$) was subtracted from the change in position of X and Y of the centrosome ($\Delta X_{Centrosome} = X_{Centrosome-t2} - X_{Centrosome-t1}$; $\Delta Y_{Centrosome} = Y_{Centrosome-t2} - Y_{Centrosome-t1}$) between each time-point to control for the movement of the centrosome resulting from the motion of the cell body ($\Delta X = \Delta X_{Cellbody} - \Delta X_{Centrosome}$; $\Delta Y = \Delta Y_{Cellbody} - \Delta Y_{Centrosome}$). Using Pythagorean theorem, net distance was calculated between time-points $d_t = \sqrt{\Delta X^2 + \Delta Y^2}$, which were then added together to calculate the total distance traveled by the centrosome $D = \sum_{t=1}^{n} d_t$; where n = final time point. For directional distance towards the cytokinetic bridge, centrin-GFP positive centrosomes were tracked from when they reach the polar ends of the daughter cells to when they reach the cytokinetic bridge using the method mentioned above. If the centrosome does not move towards the cytokinetic bridge, then the distance traveled by that centrosome is recorded as 0 $\mu m$.

### Centrosome intensity profiles

Z stacks shown are maximum projected representative cells (FIJI/ImageJ). For intensity calculations, z-stacks were sum projected, a ROI was marked around the centrosome, and mean fluorescence intensities were measured. Fluorescent intensities were calculated as mean intensity–minimum intensity. Intensities calculated were then normalized to average intensity of the parent cell population within the experiment. Three-dimensional intensity profiles were created using FIJI/ImageJ. Outliers were identified using an iterative Grubb's test with $\alpha$ = 0.05 using PRISM9 software.

### Phenotypic analysis of cells exhibiting cytokinetic defects

Human (HeLa) cells and zebrafish embryo cells (described above) were assessed for the presence of binucleated cells, represented as a percentage.

### Tracking centrosome number

Zebrafish embryo cells (described above) were assessed for abnormalities in the number of centrosomes at interphase. The number of centrosomes within each cell was counted and the percentage of cells with greater than two centrosomes was graphed using PRISM9 software.

### Statistical analysis

Unpaired, two-tailed *t* tests and one-way ANOVA were performed using PRISM9 software. **** denotes a *P*-value < 0.0001, *** *P*-value < 0.001, ** *P*-value < 0.01, * *P*-value < 0.05. For further information on detailed statistical analysis, see Table S1.

## Supplementary Information

## Acknowledgements

We thank Li-En Jao lab (UCSD) for the Tg(*pcnt*^tup2^) transgenic zebrafish lines and genotyping protocols for characterization. This work was supported by National Institutes of Health grants R35GM142963 (AE Patteson), R01GM127621 (H Hehnly) and R01GM130874 (H Hehnly), and the Syracuse University BioInspired Seed Grant (AE Patteson and H Hehnly). We thank the Blatt BioImaging Center for the use of the LSM 980 with Airyscan 2 (NIH S10 OD026946-01A1). This work was also supported by the US Army Medical Research Acquisition Activity through the FY16 Prostate Cancer Research Programs under Award no. W81XWH-20-1-0585 (H Hehnly). Opinions, interpretations, conclusions, and recommendations are those of the authors and not necessarily endorsed by the Department of Defense.

### Author Contributions

N Krishnan: conceptualization, data curation, formal analysis, validation, investigation, visualization, methodology, project administration, and writing—original draft, review, and editing.
M Swoger: resources, data curation, formal analysis, visualization, and writing—review and editing.
LI Rathbun: data curation, investigation, visualization, methodology, and writing—review and editing.
PJ Fioramonti: conceptualization, data curation, formal analysis, validation, investigation, visualization, methodology, and writing—review and editing.
J Freshour: resources, data curation, validation, investigation, methodology, and writing—review and editing.
M Bates: resources, data curation, investigation, methodology, and writing—review and editing.
AE Patteson: conceptualization, resources, formal analysis, funding acquisition, methodology, and writing—review and editing.
H Hehnly: conceptualization, resources, data curation, formal analysis, supervision, funding acquisition, validation, investigation, visualization, methodology, project administration, and writing—original draft, review, and editing.

### Conflict of Interest Statement

The authors declare that they have no conflict of interest.

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
