## [Reviewer comments · Life Science Alliance]

Rab11 endosomes and Pericentrin coordinate centrosome movement during pre-abscission in vivo.

Nikhila Krishnan, Maxx Swoger, Lindsay Rathbun, Peter Fioramonti, Judy Freshour, Michael Bates, Alison Patteson, and Heidi Hehnly

DOI: <https://doi.org/10.26508/lsa.202201362>

Corresponding author(s): Heidi Hehnly, Syracuse University

Review Timeline:	Submission Date:	2022-01-04
	Editorial Decision:	2022-01-05
	Revision Received:	2022-02-21
	Editorial Decision:	2022-02-25
	Revision Received:	2022-03-02
	Accepted:	2022-03-03

Scientific Editor: Novella Guidi

Transaction Report:

Please note that the manuscript was previously reviewed at another journal and the reports were taken into account in the decision-making process at Life Science Alliance.

Reviewer #1 Review

Comments to the Authors (Required):

This is an interesting study that investigates the movement of centrosomes during division using *in vitro* and *in vivo* (zebrafish) models. The migration of some centrosomes to the cytokinetic bridge is an interesting phenomenon that has not been extensively studied, thus, the manuscript does deal with very interesting topic. One issue is, however, that many experiments are confirmatory and describes phenomenon already defined by several labs using several experimental models (although not in zebrafish). For example, centrosomal migration toward bridge in HeLa cells has been described before. Similarly, there are numerous studies describing the dynamics of Rab11 and FIP3 during cell division. Finally, centrosomal migration toward apical side (where cytokinetic bridge is localized) has also been described in *Xenopus* and in MDCK cells. Thus, first three figures have little novelty and simply recapitulates what is already known. Even bigger issue is the question of why centrosomes move toward cytokinetic bridge? While this phenomenon has been described over a decade ago, we still do not know whether this movement has any function. Indeed, this movement is not observed in all cell lines. Additionally, by the time cell gets to late telophase and partially flattens out, often (even in HeLa cells) centrosomes moves away from the cytokinetic bridge, thus, unlikely that this movement play a role in abscission regulation. Similarly, centrosomes usually move to the apical side to establish cilia after completing abscission, thus, it is unclear why this movement toward the bridge would play a role in ciliation. Unfortunately authors do not investigate the function of centrosome movement, but simply provides descriptonal analysis of centrosomal movement during mitosis. Without clear function for this movement, the work will have only marginal interest to wider cell biology community. Finally, manuscript also has some technical issues (like generation of Rab11-null cells) that need to be addressed (see below).

1) Figure 2. FIP3 and Rab11 dynamics during cell division has been extensively studied by quite a few laboratories. It appears that late during telophase (just before abscission) Rab11 and FIP3 leave centrosome and relocalize to the abscission site. Do authors observe that in their studies (in HeLa cells)? How close centrosomes actually get to the bridge? How close do they need to get for authors to classify them as "daughter cell with bridge directed centrosome movement".

2) Figure 3G. The statement that "strikingly, at the oldest centrosome Rab11 accumulated around the mother centriole" needs to be backed up with quantifications. Based on the image shown, authors could easily measure the distances between endosomes and either mother or daughter centriole. Additionally, it has been reported by a few labs that rab11 endosomes associate with distal appendages, thus, this finding is not really that surprising or novel.

3) Figure 4. I could not find much info about how Rab11-null cells were generated. Are they lacking both Rab11a and Rab11b? Have they been genotyped and western blotted to confirm that both Rab11 isoforms have been knocked out? Considering that Rab11 isoforms have a lot of redundancy, it would be key to demonstrate that.

4) Figure 5. Most cells shown in this figure are at early telophase as can be determined by the nucleus shape and compactness of central spindle microtubules. Since authors are trying to prove the connection between centrosomes and abscission, the need to show cells that are in late telophase, just before entering abscission.

Reviewer #2 Review

Comments to the Authors (Required):

This manuscript describes the migration of centrosomes towards the cytokinetic bridge during cell division in zebrafish embryos (Kupfer vesicles) and HeLa cells. Rab11 is known to associate with mother centriole appendage proteins during interphase (Hehnly 2012), and Rab11 GTPase cycling appears to be needed for this type of directed centrosome movement, possibly by promoting deposition of Pericentrin onto the centrosomes. Reduction of Pericentrin levels or disruption of Rab11-endosome function both reduced directed centrosome movement and promoted failure of normal abscission.

Overall, the study is potentially interesting, and uses multiple models, but is mostly descriptive and the first half is largely a recapitulation of known processes in *Drosophila* and mammalian cells that have been reported by multiple laboratories. One concern is that the movement of centrosomes towards the cytokinetic bridge is not universal and does not always occur even in HeLa cells or in the zebrafish embryo (no movement in about 30% of divisions). Therefore, this process cannot be considered essential for normal cell division. While optogenetic clustering of Rab11 causes a significant increase in binucleation and supernumerary centrosomes this effect might be causally independent of the centrosome movement towards the cytokinetic bridge.

There are also some technical issues:

1) From the methods section it appears that only Rab11A was deleted, so presumably Rab11B is still present in the cells and they are not "Rab11 null". Were the CRISPR-edited cells clonal or not?

2) ON page 11 it states that the mCherry-Rab11 expression levels were similar to endogenous Rab11 in control cells, but this is clearly untrue - from the immunoblot in Fig S3D the Cherry-Rab11 expression appears several fold higher than endogenous expression (lane 1) and this will perturb FRAP experiments because not all of the over-expressed Rab11 will likely be correctly localized/interacting appropriately. In general, over-expressed proteins show a significantly larger mobile fraction than endogenously tagged proteins. Also, the level of mCh-Rab11(Q70L) is lower than that of the WT which might account for the small difference seen in % mobile fraction, so these data are not convincing.

3) Likewise, the FRAP data in Fig 3E, F are not convincing. The effect size is very small, and the "immobile" fraction is not really immobile, just recovering more slowly. Also Fig 3G is based on a single cell image with no quantification. Also Fig 1E and 4C need statistical analysis.

Overall the study, while potentially interesting, would need significantly further insight into the mechanistic link between centrosome movement and abscission, or between Rab11 and centrosomal Pericentrin levels to meet the criteria for acceptance by this journal.

January 5, 2022

Re: Life Science Alliance manuscript #LSA-2022-01362-T

Dr. Heidi Hehnly
Syracuse University
Biology
Life Sciences Complex
107 College Pl
Syracuse, New York 13210

Dear Dr. Hehnly,

Thank you for submitting your manuscript entitled "Rab11 endosomes coordinate centrosome movement during pre-abscission" to Life Science Alliance. The manuscript was previously assessed at another journal by expert reviewers and then transferred to LSA. We invite you to submit a revised manuscript addressing the following revisions:

- Address Reviewer 1's technical issues excluding the request to investigate the function of centrosome movement.
- Address Reviewer 2's technical issues, excluding the request to explore mechanistic link between centrosome movement and abscission or between Rab11 and centrosomal Pericentrin levels.

Thank you for this interesting contribution to Life Science Alliance. We are looking forward to receiving your revised manuscript.

Sincerely,

-- By submitting a revision, you attest that you are aware of our payment policies found here: <https://www.life-science->

alliance.org/copyright-license-fee

B. MANUSCRIPT ORGANIZATION AND FORMATTING:

We thank Life Science Alliance for inviting us to submit a revised manuscript with the following revisions addressed:

*"- Address Reviewer 1's technical issues excluding the request to investigate the function of centrosome movement.
- Address Reviewer 2's technical issues, excluding the request to explore mechanistic link between centrosome movement and abscission or between Rab11 and centrosomal Pericentrin levels."*

We have taken the editorial response team's recommendation (in green above) to address only the technical issues pointed out by the reviewers and modified the manuscript accordingly. We have highlighted these technical concerns from reviewer comments in green font and answered them below in blue font. Corresponding changes within the main manuscript is also noted in blue font. New figures that have been added in response to the reviewer concerns include: Figs 1E, 3E, 3F, 3H, 3I, 4C, 6C; Supplementary Figs S1C, S1D, S1E, S3I, S4F, S4G.

Reviewer #1:

This is an interesting study that investigates the movement of centrosomes during division using in vitro and in vivo (zebrafish) models. The migration of some centrosomes to the cytokinetic bridge is an interesting phenomenon that has not been extensively studied, thus, the manuscript does deal with very interesting topic. One issue is, however, that many experiments are confirmatory and describes phenomenon already defined by several labs using several experimental models (although not in zebrafish). For example, centrosomal migration toward bridge in HeLa cells has been described before. Similarly, there are numerous studies describing the dynamics of Rab11 and FIP3 during cell division. Finally, centrosomal migration toward apical side (where cytokinetic bridge is localized) has also been described in Xenopus and in MDCK cells. Thus, first three figures have little novelty and simply recapitulates what is already known.

Even bigger issue is the question of why centrosomes move toward cytokinetic bridge? While this phenomenon has been described over a decade ago, we still do not know whether this movement has any function. Indeed, this movement is not observed in all cell lines. Additionally, by the time cell gets to late telophase and partially flattens out, often (even in HeLa cells) centrosomes moves away from the cytokinetic bridge, thus, unlikely that this movement play a role in abscission regulation. Similarly, centrosomes usually move to the apical side to establish cilia after completing abscission, thus, it is unclear why this movement toward the bridge would play a role in ciliation.

Unfortunately authors do not investigate the function of centrosome movement, but simply provides descriptive analysis of centrosomal movement during mitosis. Without clear function for this movement, the work will have only marginal interest to wider cell biology community. Finally, manuscript also has some technical issues (like generation of Rab11-null cells) that need to be addressed (see below).

Technical Issues:

1. *Figure 2. FIP3 and Rab11 dynamics during cell division has been extensively studied by quite a few laboratories. It appears that late during telophase (just before abscission) Rab11 and FIP3 leave centrosome and relocalize to the abscission site. Do authors observe that in their studies (in HeLa cells)? How close centrosomes actually get to the bridge? How close do they need to get for authors to classify them as "daughter cell with bridge directed centrosome movement"*

For the question “Rab11 and FIP3 leave centrosome and relocalize to the abscission site. Do authors observe that in their studies (in HeLa cells)?” To test this we performed a photoconversion study of Dendra-Rab11 at the centrosome, and found that the photoconverted centrosome population of Rab11 was the population that moved towards the bridge (Fig 2B,C). This population of Rab11 (Fig 2B,C) and GFP-FIP3 (Fig 2A) moves with the centrosome as it moves towards the bridge and then leaves the centrosome where it can then move into the bridge (Fig 2A-C, Video S2).

In response to “How close centrosomes actually get to the bridge?” we calculated the distance of the centrosome to the bridge center when the centrosome was at its closest point to the bridge in our control HeLa live-cell imaging data sets. We presented these studies as a histogram of centrosome distances from the bridge under control conditions (*new* Fig S1E). We see a range of distances due to the bridge and cells having a range of lengths and sizes respectively, but consistently we do find that the centrosome at least moves within 4 μm to the midway point of the cytokinetic bridge (example in *new* Fig S1C, quantification in Fig S1E). We find that the centrosome rarely goes directly into the cytokinetic bridge in our live-cell imaging data sets, but goes adjacent to the bridge. With this and our photoconversion studies of Rab11 at centrosomes (Fig 2B-C) we ascertained that a Rab11 centrosome population is what moves into the bridge as the centrosome is moving towards the bridge.

For the question “How close do they need to get for the authors to classify them as daughter cell with bridge directed centrosome movement” we have created a diagram of the quantification methodology used in *new* Fig S1D. In short, we monitored in live-cell movies the number of daughter cells where the centrosomes moved from the polar end of the cell across the midpoint of the daughter cell (refer to movements in Fig S1A and B). The midpoint or centroid position is depicted with a green dot in Fig S1C and S1D. If the centrosome had bridge-directed movement past this point, then it was determined as a daughter cell with bridge directed centrosome movement. This is now described in the methods and result sections.

2. *Figure 3G. The statement that "strikingly, at the oldest centrosome Rab11 accumulated around the mother centriole" needs to be backed up with quantifications. Based on the image shown, authors could easily measure the*

distances between endosomes and either mother or daughter centriole. Additionally, it has been reported by a few labs that rab11 endosomes associate with distal appendages, thus, this finding is not really that surprising or novel.

We thank the reviewer for their suggestion, and while it was difficult for us to measure distances, we decided to measure the area of endosomes that touched or overlapped in signal to the mother or daughter centriole. We found a significantly increased area of endosomes that interacted with the mother compared to the daughter. Refer to new Fig 3H with added example and model of the endosome area that is calculated in new Fig 3I across expanded samples.

In response to “it has been reported by a few labs that rab11 endosomes associate with distal appendages, thus, this finding is not really that surprising or novel.” We kindly argue that our studies were one of the first that showed that Rab11 interacts with sub-distal appendages in non-ciliated interphase cells and our findings by EM did not find an association with distal appendages (Hehnly et al., 2012). An additional report demonstrated that Rab8, and potentially the Rab11-Rab8 cascade, has an association with distal appendages during ciliogenesis (Schmidt et al., 2012) suggesting that at different cell cycle stages different flavors of Rab11 or RE interactions may occur with the centrosome. It has yet to be reported if there is a relationship between Rab11-endosomes and the mother centriole during mitotic stages and these previous studies did not demonstrate that this interaction occurs during telophase to pre-abscission, nor that centrosomes and Rab11/FIP3 co-localize together during the duration of telophase to pre-abscission as the centrosome moves towards the bridge. There are ample reports that Rab11 is involved in abscission (Wilson et al., 2005; Schiel et al., 2012), but the relationship of Rab11 with the centrosome during this time has not been investigated. Thus, our study is confirming and expanding on the initial foundational studies where we suggest an importance for Rab11-endosome interaction with the centrosome during multiple cell cycle stages. We have clarified this as discussion points in the discussion section.

3. Figure 4. I could not find much info about how Rab11-null cells were generated. Are they lacking both Rab11a and Rab11b? Have they been genotyped and western blotted to confirm that both Rab11 isoforms have been knocked out? Considering that Rab11 isoforms have a lot of redundancy, it would be key to demonstrate that.

The Rab11-null cell lines were created using Rab11a CRISPR vector and Rab11a HDR Vector (for catalog # see Key resource table) and characterization done in (Rathbun et al., 2020). Western blot analysis to confirm Rab11a loss used an antibody that detects both Rab11a and Rab11b (Fig S3D, Key Resource Table). The Western blot in Fig S3D demonstrates loss of Rab11 signal in our Rab11-null cells compared to controls suggesting that both Rab11a and Rab11b are not present or Rab11b is present at low and undetectable amounts. We have clarified this in the main text and in the methods section. We also have clarified that we are rescuing the binucleated phenotype we find with Rab11a null cells with mCherry-Rab11a in the main text on page 10.

4. *Figure 5. Most cells shown in this figure are at early telophase as can be determined by the nucleus shape and compactness of central spindle microtubules. Since authors are trying to prove the connection between centrosomes and abscission, the need to show cells that are in late telophase, just before entering abscission.*

In Fig 5 we are using early abscission/telophase cells where all the centrosomes are on the polar ends before they move to the bridge. We did this intentionally to address the status of centrosome integrity with Rab11 loss or rescue conditions. The reason for doing this is that we can directly compare the centrosome when they are in the same position across conditions, for instance in Rab11 null conditions the centrosome (in most cases) stays at the polar end of the cell, where in control conditions it moves towards the bridge at the time of late abscission (Fig 4). In response to your concerns, we identified and confirmed that Rab11-null late abscission cells had a significant reduction in pericentrin at the centrosome compared to control cells (*new* Fig S4F and S4G). The aim of Fig 5 was not to prove the connection between centrosomes and abscission but to demonstrate the connection between Rab11 and centrosome integrity where we found that loss of Rab11 function causes a reduction in Pericentrin at the centrosome. We have clarified this in the text. Fig 6 is where we are trying to demonstrate a connection between a loss in centrosome function (decreases in Pericentrin) results in centrosome defects in moving towards the bridge with associated abscission defects.

Reviewer #2:

This manuscript describes the migration of centrosomes towards the cytokinetic bridge during cell division in zebrafish embryos (Kupfer vesicles) and HeLa cells. Rab11 is known to associate with mother centriole appendage proteins during interphase (Hehnlly 2012), and Rab11 GTPase cycling appears to be needed for this type of directed centrosome movement, possibly by promoting deposition of Pericentrin onto the centrosomes. Reduction of Pericentrin levels or disruption of Rab11-endosome function both reduced directed centrosome movement and promoted failure of normal abscission.

Overall, the study is potentially interesting, and uses multiple models, but is mostly descriptive and the first half is largely a recapitulation of known processes in Drosophila and mammalian cells that have been reported by multiple laboratories.

One concern is that the movement of centrosomes towards the cytokinetic bridge is not universal and does not always occur even in HeLa cells or in the zebrafish embryo (no movement in about 30% of divisions). Therefore, this process cannot be considered essential for normal cell division. While optogenetic clustering of Rab11 causes a significant increase in binucleation and supernumerary centrosomes this effect might be causally independent of the centrosome movement towards the cytokinetic bridge.

Technical Issues:

1. *From the methods section it appears that only Rab11A was deleted, so presumably Rab11B is still present in the cells and they are not "Rab11 null"? Were the CRISPR-edited cells clonal or not?*

This was a similar concern with Reviewer 1, Question 3. We have now clarified our text and in short, we used a Rab11a CRISPR vector and Rab11a HDR Vector (for catalog # see Key Resource Table). Western blot analysis was used to confirm Rab11a loss where we used an antibody that detects both Rab11a and Rab11b (Key Resource Table). The Western blot in Fig S3D demonstrates loss of Rab11 signal in our Rab11-null cells compared to controls suggesting that both Rab11a and Rab11b isoforms are not present or Rab11b is present at low and undetectable amounts. We have clarified this in the main text and in methods. We also have clarified that we are rescuing the binucleated phenotype we find with Rab11a null cells with mCh-Rab11a in the main text on page 11.

- 2. ON page 11 it states that the mCherry-Rab11 expression levels were similar to endogenous Rab11 in control cells, but this is clearly untrue - from the immunoblot in Fig S3D the Cherry-Rab11 expression appears several fold higher than endogenous expression (lane 1) and this will perturb FRAP experiments because not all of the over-expressed Rab11 will likely be correctly localized/interacting appropriately. In general, over-expressed proteins show a significantly larger mobile fraction than endogenously tagged proteins. Also, the level of mCh-Rab11(Q70L) is lower than that of the WT which might account for the small difference seen in % mobile fraction, so these data are not convincing.*

We have amended the statement on Page 12 to address the relative abundance of Rab11 rescues. While this overexpression is a concern with FRAP experiments, our experiments aren't necessarily commenting on the endogenous behavior of Rab11, but that mCh-Rab11 in the same pre-abscising cell has different kinetics between daughter cells that is associated with centrosome age. We have clarified the text and figure to highlight this, where we now include 3 examples of single cells with fitted FRAP curves of mCh-Rab11 at the oldest and youngest centrosome (*new* Fig 3E). We then present the T1/2 and mobile fractions of mCh-Rab11 from the 3 cells in Fig 3E as a ratio of mCh-Rab11 measurements at oldest centrosome over youngest centrosome. If there was no difference in the mobile fraction or T1/2 then the ratio should be around 1. For the mobile fraction we find a mean of 1.8 ± 0.2 that significantly deviates from 1, and for the T1/2 we find a mean ratio of 1.2 ± 0.2 that does not significantly deviate from 1 (*new* Fig 3F). For the mCh-Rab11 and mCh-Rab11(Q70L) data we carefully used cells that fell within a range of similar fluorescent intensities (see fluorescent quantification of cells included for FRAP analysis now included in Fig S3I) so that a comparison can be made between mCh-Rab11 and mCh-Rab11(Q70L) dynamics at the centrosome in cells at pre-abscission when the centrosomes are still at the polar ends of the cell. This has now been clarified and explanation expanded upon in the text.

- 3. Likewise, the FRAP data in Fig 3E, F are not convincing. The effect size is very small, and the "immobile" fraction is not really immobile, just recovering more slowly. Also Fig 3G is based on a single cell image with no quantification.*

To clarify our FRAP experiments and to demonstrate the difference in mCh-Rab11 kinetics between two daughter cells that derived from the same mother, we have included 3 example traces of daughter cells derived from the same mother cell in pre-

abscission with a fitted FRAP curve of mCh-Rab11 from the oldest and youngest centrosome (*new* Fig 3E). As suggested, we now include the T1/2 and “mobile” fractions in Fig 3F. To highlight the difference in mCh-Rab11 dynamics between the oldest and youngest centrosomes between the two daughter cells we took a ratio of the mobile fraction and T1/2 of the oldest/youngest centrosome (*new* Fig 3F). In Fig 3E, while there is variation in the degree of mobility between individual cells, we find that the oldest centrosome consistently has an elevated mobile fraction when compared to the younger centrosome in all three cases (Fig 3E and 3F). and the ratio of that mobility of oldest over youngest significantly deviates from 1. The variation in the degree of the mobile fraction differences between the oldest and youngest centrosome may be due to inconsistencies in expression or variations in pre-abscission stages that should be taken under consideration. With this in mind, our findings suggest that mCh-Rab11 has an elevated mobile fraction at the oldest centrosome compared to the youngest. This has now been discussed in the main text.

In response to “Figure 3G is based on a single cell image with no quantification” we have now included quantification in *new* Fig 3I. We measured the area of expanded endosomes that touched or overlapped in signal to the expanded mother or daughter centriole. We found a significantly increased area of endosomes that interacted with the mother compared to the daughter. Refer to *new* Fig 3H that has an added example and model of the endosome area that is calculated in *new* Fig 3I across expanded samples.

4. Also Fig 1E and 4C need statistical analysis.

Since we are characterizing the total percent of cells that display a phenotypic significance in Fig 1E and 4C traditionally studies do not perform stats for this sort of data representation (refer to studies (Lu et al., 2015; Liu et al., 2017)). We wanted to present graphs like this to show the relationship in the percentage of daughter cells with bridge directed centrosome movement across cell types, model systems and centrosome markers to highlight general similarities and differences. To present the studies where statistical analysis can be employed as Reviewer 2 suggests, we calculated whether both daughter cell centrosomes (2 centrosomes), only one daughter cell centrosome (1 centrosome), or neither centrosome (0 centrosome) moved from the polar ends of the daughter cells across the cell centroid position (modeled in *new* Fig S1D) towards the cytokinetic bridge (Fig S1C-S1E). The median (orange dashed line) and quartiles (dark dotted lines) are displayed in a violin plot, refer to *new* Fig 1E, 4C and 6C. Refer to Table 1 for detailed statistical results for these studies.

Overall the study, while potentially interesting, would need significantly further insight into the mechanistic link between centrosome movement and abscission, or between Rab11 and centrosomal Pericentrin levels to meet the criteria for acceptance by this journal.

References:

Liu, Y., D.S. Sepich, and L. Solnica-Krezel. 2017. Stat3/Cdc25a-dependent cell proliferation promotes embryonic axis extension during zebrafish gastrulation. 13. 1–32 pp.

- Lu, Q., C. Insinna, C. Ott, J. Stauffer, P.A. Pintado, J. Rahajeng, U. Baxa, V. Walia, A. Cuenca, Y.S. Hwang, I.O. Daar, S. Lopes, J. Lippincott-Schwartz, P.K. Jackson, S. Caplan, and C.J. Westlake. 2015. Early steps in primary cilium assembly require EHD1/EHD3-dependent ciliary vesicle formation. *Nat. Cell Biol.* 17:228–240. doi:10.1038/ncb3109.
- Rathbun, L.I., E.G. Colicino, J. Manikas, J. O’Connell, N. Krishnan, N.S. Reilly, S. Coyne, G. Erdemci-Tandogan, A. Garrastegui, J. Freshour, P. Santra, M.L. Manning, J.D. Amack, and H. Hehny. 2020. Cytokinetic bridge triggers de novo lumen formation in vivo. *Nat. Commun.* 11:1269. doi:10.1038/s41467-020-15002-8.
- Schiel, J.A., G.C. Simon, C. Zaharris, J. Weisz, D. Castle, C.C. Wu, and R. Prekeris. 2012. FIP3-endosome-dependent formation of the secondary ingression mediates ESCRT-III recruitment during cytokinesis. *Nat. Cell Biol.* 14:1068–1078. doi:10.1038/ncb2577.
- Schmidt, K.N., S. Kuhns, A. Neuner, B. Hub, H. Zentgraf, and G. Pereira. 2012. Cep164 mediates vesicular docking to the mother centriole during early steps of ciliogenesis. *J. Cell Biol.* 199:1083–1101. doi:10.1083/jcb.201202126.
- Wilson, G.M., A.B. Fielding, G.C. Simon, X. Yu, P.D. Andrews, R.S. Haines, A.M. Frey, A.A. Peden, G.W. Gould, and R. Prekeris. 2005. The FIP3-Rab11 protein complex regulates recycling endosome targeting to the cleavage furrow during late cytokinesis. *Mol. Biol. Cell.* 16:849–860. doi:10.1091/mbc.E04-10-0927.

February 25, 2022

RE: Life Science Alliance Manuscript #LSA-2022-01362-TR

Dr. Heidi Hehnlly
Syracuse University
Biology
Life Sciences Complex
107 College Pl
Syracuse, New York 13210

Dear Dr. Hehnlly,

Thank you for submitting your revised manuscript entitled "Rab11 endosomes and Pericentrin coordinate centrosome movement during pre-abscission in vivo.". We would be happy to publish your paper in Life Science Alliance pending final revisions necessary to meet our formatting guidelines.

- please upload your main and supplementary figures as single files;
- please add the Twitter handle of your host institute/organization as well as your own or/and one of the authors in our system
- please use the [10 author names, et al.] format in your references (i.e. limit the author names to the first 10)

A. FINAL FILES:

B. MANUSCRIPT ORGANIZATION AND FORMATTING:

**Submission of a paper that does not conform to Life Science Alliance guidelines will delay the acceptance of your

manuscript.**

The license to publish form must be signed before your manuscript can be sent to production. A link to the electronic license to publish form will be sent to the corresponding author only. Please take a moment to check your funder requirements.

Sincerely,

March 3, 2022

RE: Life Science Alliance Manuscript #LSA-2022-01362-TRR

Dr. Heidi Hehnly
Syracuse University
Biology
Life Sciences Complex
107 College Pl
Syracuse, New York 13210

Dear Dr. Hehnly,

Thank you for submitting your Research Article entitled "Rab11 endosomes and Pericentrin coordinate centrosome movement during pre-abscission in vivo.". It is a pleasure to let you know that your manuscript is now accepted for publication in Life Science Alliance. Congratulations on this interesting work.

DISTRIBUTION OF MATERIALS:

Again, congratulations on a very nice paper. I hope you found the review process to be constructive and are pleased with how the manuscript was handled editorially. We look forward to future exciting submissions from your lab.

Sincerely,
